# Distinct echinocandin responses of *Candida albicans* and *Candida auris* cell walls revealed by solid-state NMR

Malitha C. Dickwella Widanage[1,7,8], Kalpana Singh[1,8], Jizhou Li[2,3,8], Jayasubba Reddy Yarava [1], Faith J. Scott[4], Yifan Xu[1], Neil A. R. Gow [5], Frederic Mentink-Vigier [4], Ping Wang [6], Frederic Lamoth [2,3] ✉ & Tuo Wang [1] ✉

Invasive candidiasis affects 1.6 million people annually, with high mortality among immunocompromised and hospitalized patients. Echinocandins are frontline antifungals, but rising resistance limits their efficacy. Here, we show that *Candida albicans* and multidrug-resistant *Candida auris* share a conserved cell wall architecture yet differ markedly in their adaptive responses to echinocandins. Solid-state NMR reveals that both species possess a rigid inner layer of tightly associated chitin microfibrils and β−1,3-glucans, supported by a flexible matrix of β−1,6-glucans and additional β−1,3-glucans. Outer mannan fibrils rely on α−1,2-linked sidechains to maintain contact with the inner wall. In both species, caspofungin rigidifies β−1,6-glucans and mannan sidechains and reduces water permeability during β−1,3-glucan depletion; however, *C. albicans* undergoes wall thickening and alterations in chitin and glucan dynamics, whereas *C. auris* maintains integrity through β−1,6-glucan upregulation. Deletion of *KRE6a*, which encodes β−1,6-glucan synthase, reduces echinocandin susceptibility in *C. auris*, further highlighting β−1,6-glucan's critical role in adaptive remodeling.

*C*andida species cause superficial and systemic infections that represent significant challenges for healthcare systems due to high morbidity and mortality, particularly in immunocompromised individuals[1,2]. Recent estimates indicate that 700,000 to 1.6 million individuals suffer from invasive candidiasis, including bloodstream infections, with a 40−75% mortality rate across various clinical settings[3,4]. Among these species, *C. albicans* is the most prevalent[5], while *C. auris*, a multidrug-resistant pathogen that recently emerged simultaneously on three continents, has also been classified as a critical

priority pathogen by the World Health Organization[6–8]. Most clinical isolates of *C. auris* exhibit acquired resistance to one or more, or in some cases all, of the antifungal drugs commonly used to treat these infections[9]. Both *C. albicans* and *C. auris* have been implicated in severe coinfections during the COVID-19 pandemic[10], with outbreaks persisting in hospital settings despite decontamination efforts, especially in long-term care facilities and COVID-19 treatment units[11–15].

Currently, only three classes of antifungal drugs are available to treat invasive candidiasis−azoles, polyenes, and echinocandins,

[1]Department of Chemistry, Michigan State University, East Lansing, MI, USA. [2]Institute of Microbiology and Service of Infectious Diseases, Lausanne University Hospital and University of Lausanne, Lausanne, Switzerland. [3]Infectious Diseases Service, Lausanne University Hospital and University of Lausanne, Lausanne, Switzerland. [4]National High Magnetic Field Laboratory, Florida State University, Tallahassee, FL, USA. [5]Medical Research Council Centre for Medical Mycology at the University of Exeter, University of Exeter, Geoffrey Pope Building, Stocker Road, Exeter EX4 4QD, UK. [6]Departments of Microbiology, Immunology and Parasitology, Louisiana State University Health Sciences Center, New Orleans, LA, USA. [7]Present address: National Renewable Energy Laboratory, Golden, CO, USA. [8]These authors contributed equally: Malitha C. Dickwella Widanage, Kalpana Singh, Jizhou Li. ✉e-mail: frederic.lamoth@chuv.ch; wangtuo1@msu.edu

although new options, such as the next-generation echinocandin called rezafungin and the first-in-class antifungals ibrexafungerp and fosmanogepix, are in development[16–19]. Echinocandins, which inhibit the synthesis of β−1,3-glucan, an essential polysaccharide in the fungal cell wall, are recommended as the first-line treatment of *Candida* infections and have been widely used in clinical settings[20,21]. However, echinocandin treatment often results only in partial inhibition of β−1,3-glucan synthesis[22–24]. Resistance to echinocandins has been widely reported in *C. auris* strains, where it occurs naturally, and in *C. albicans*, where the incidence has been increasing primarily due to mutations in two hotspot regions of β−1,3-glucan synthase Fks subunit[25,26]. These effects have been associated with cell wall adaptation to echinocandins, characterized by increased levels and exposure of chitin following the depletion of a substantial portion of β-glucan; however, our understanding of echinocandin-induced remodeling mechanism of *Candida* cell wall architecture remains limited[24,27–29].

The fungal cell wall plays a central role in immune recognition and response by acting as a barrier to external stress, facilitating communication with the environment, and contributing to virulence[4,30,31]. Due to its structural differences from human cells, the fungal cell wall is also an important target for novel antifungal therapies[32,33]. *C. albicans* has a well-characterized cell wall composed of an inner layer of chitin, β−1,3-glucan, and β−1,6-glucan, and an outer layer composed of proteins modified with *N*- and *O*-linked mannans and phosphomannans, called mannoproteins[34,35]. These mannan components, which often account for 30–40% of the cell wall polysaccharides, are linked to the β−1,3-glucan-chitin network either via the β−1,6-glucan moiety or directly[36]. Much of this structural information is derived from studies focused on *C. albicans*, often in comparison with the model yeast *Saccharomyces cerevisiae*.

*C. auris* has recently emerged as a significant threat to hospitalized patients, and its cell wall structure has been understudied[6]. *C. auris* is classified into five genetically distinct clades (I–V) with geographically specific distributions across South Asia, East Asia, Africa, South America, and Iran, and a newly proposed Clade VI identified in the Indomalayan region[37–39]. These clades exhibit different patterns of antifungal resistance and mating types, with Clades I, III, IV, and V associated with invasive infections, while Clade II is predominantly linked to ear infections[40–42]. Despite regional predominance, infections from multiple clades have been transmitted to the United States, Canada, and the United Kingdom[43,44]. Almost all *C. auris* isolates are resistant to fluconazole, over half to voriconazole, one-third to amphotericin B, and some are resistant to all major classes of antifungal agents, including echinocandins[45–48]. The high mortality rates and limited treatment options underscore the urgent need for new antifungal therapies. Due to its distinction in drug resistance characteristics from other *Candida* species, an in-depth examination of *C. auris* cell wall structure is warranted[49].

To bridge such a knowledge gap, here we employ multidimensional ¹³C/¹⁵N and ¹H-detected solid-state NMR and dynamic nuclear polarization (DNP) to characterize the structure of polysaccharides within intact ¹³C, ¹⁵N-labeled cells *C. auris*, and determine their distribution and physical organization within dynamically distinct domains of the cell wall, in comparison to of *C. albicans*. Some of these solid-state NMR techniques have been selectively applied and tailored−depending on the purpose of each study−to investigations of the morphotype-dependent cell wall structure and stress responses in *Aspergillus* (including *A. fumigatus*, *A. nidulans*, and *A. sydowii*)[50–53], *Cryptococcus*[54–56], *Schizophyllum*[57,58], and more recently, *Neurospora*, *Mucor* and *Rhizopus* species[59,60]. By modifying these techniques, the current study reveals that, while both *C. albicans* and *C. auris* share a similar global cell wall composition different from that of other fungi, such as *A. fumigatus*, they exhibited markedly distinctive cell wall structural architectures in response to caspofungin exposure. These findings will help to guide future examination of antifungal effectiveness against different *Candida* species by resolving differences in structural and mechanistic regulatory mechanisms of the cell walls, despite a similarity in their overall polysaccharide compositions.

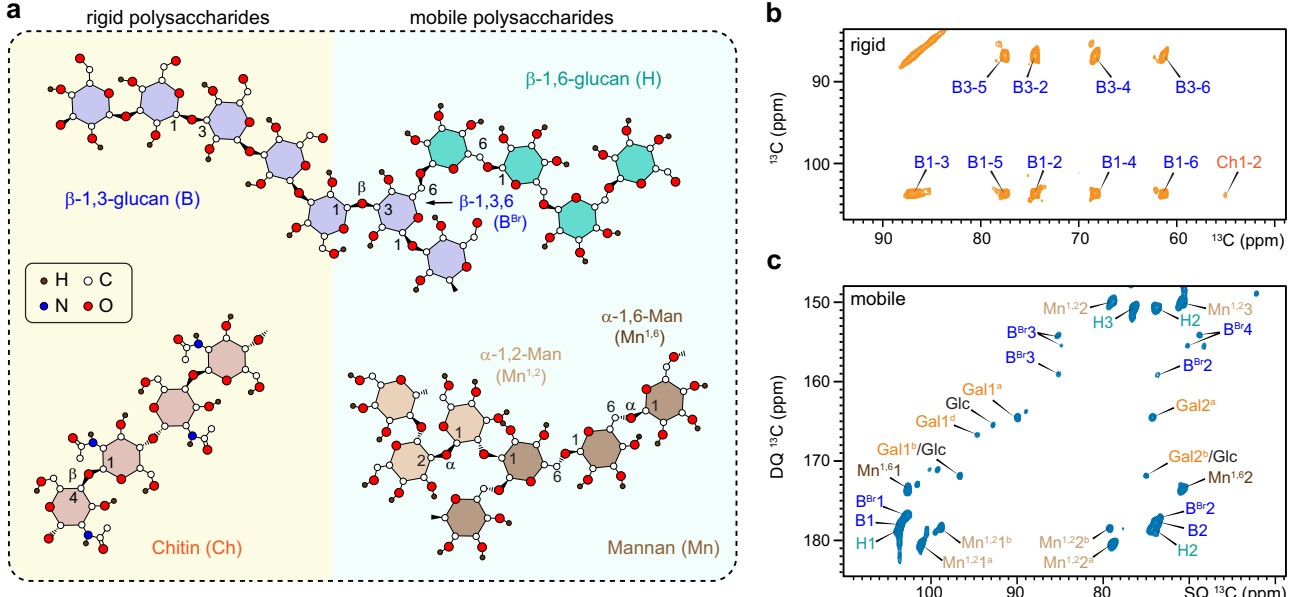

**Fig. 1 | Dynamically distinct carbohydrate domains in *Candida* species.**
**a** Simplified structures of *Candida* cell wall polysaccharides within mobile and rigid phases as highlighted in pale yellow and cyan background colors, respectively. The color code and NMR abbreviations for different carbohydrates are provided. H: β-1,6-glucan (cyan); B: β−1,3-glucan (blue); B^Br: β−1,3,6-linked glucose unit; Mn^1,2: α−1,2-linked mannose (light brown); Mn^1,6: α−1,6-linked mannose (brown); Ch: chitin (orange). Filled black circles: H atoms; open circles: C atoms; blue circles: N atoms; red circles: O atoms. **b** 2D ¹³C-¹³C 53-ms CORD spectra detecting rigid cell wall polysaccharides in *C. albicans*. **c** Mobile molecules of *C. albicans* detected through 2D ¹³C DP J-INADEQUATE spectrum. Additional abbreviations: Gal: galactose; Glc: glucose.

## Results

### *Candida* cell wall features a simple two-component rigid core embedded in soft matrix

*Candida* species contain long β−1,3-glucans and short β−1,6-glucans, with the latter potentially serving as a bridge between the former and cell wall proteins (Fig. 1a)[61]. β−1,6-glucan is primarily linear and contains short side chains of β−1,3-glucoside or laminaribioside residues[36,62]. Solid-state NMR spectra of intact *C. albicans* cells revealed that β−1,3-glucans (B), alongside a lower content of chitin (Ch), are the major component of the rigid fraction of the cell wall, as detected by 2D $^{13}$C-$^{13}$C CORD correlation spectra (Fig. 1b). These components exhibit reduced molecular mobility due to the formation of ordered or semi-crystalline domains, or tightly packed aggregates that enhance the stiffness of the cell wall[63,64]. Meanwhile, the mobile carbohydrates in *C. albicans* detected using a DP-based refocused J-INADEQUATE spectrum primarily correspond to components of the β-glucan matrix, including β-1,3-linkages (B) in the main chain, β-1,3,6-linkages (B$^{Br}$) at branching points, and β-1,6-linkages (H) in the sidechains (Fig. 1c). Positioned within the soft mesh and spatially distinct from the rigid core, these mobile molecules generally remain loosely associated and well-hydrated[63,64]. These mobile domains regulate water accessibility and influence the diffusion of soluble molecules, including antifungals, while rigid, often partially dehydrated structural domains serve as a physical barrier, reducing overall permeability. Recent observations suggest that the balance between rigid structural components and mobile, water-accessible regions allows fungi to fine-tune cell wall permeability, aiding adaptive survival[50,65].

In addition, within the mobile domain, two types of β-1,3,6-linked glucopyranose units were identified, as indicated by the splitting of the B$^{Br}$3 and B$^{Br}$4 peaks. This likely arises from these units contributing to two distinct structural environments: one where β-1,3-glucan branches to another β-1,3-glucan, forming branched β-1,3/1,6-glucan, and the other where β-1,3-glucan branches to a linear β-1,6-glucan. Additionally, signals from α−1,2- and α−1,6-linked mannose residues (Mn$^{1,2}$ and Mn$^{1,6}$) in mannan fibrils and weak signals from galactose (Gal) residues were observed in the mobile phase. The $^{13}$C chemical shifts of these molecules were documented in Supplementary Table 1.

It should be noted that distribution of these carbohydrate molecules across dynamically distinct phases, as demonstrated in the SC5314 strain, was consistently observed in the spectra of other *C. albicans* strains, including JKC1450, JCK1583, JKC302, as well as BS1 strains (Supplementary Fig. 1)[66]. These findings demonstrate that *C. albicans* cell walls share a relatively simple structure, with the rigid structural framework containing only chitin microfibrils and stiff linear β-1,3-glucan backbones, which may primarily adopt the triple helix conformation[67–69], within the inner layer. These rigid components are supported by more flexible β-1,6-glucans and a small portion of branched β-1,3-glucans of the glucan complex. Both β-1,6-glucans and branched β-1,3-glucans have a branching topology and may exhibit a reduced degree of intermolecular association compared to chitin microfibrils and β-1,3-glucan triple helices. Mannan forms fibrillar structures that particularly dominate the outer layer of the cell wall, and our findings indicate that these fibrils are dynamically fluctuating, in contrast to the chitin microfibrils that exhibit partial structural order and crystallinity, as well as stiffness.

### *C. albicans* and *C. auris* share similar overall cell wall composition

To compare the cell walls of *C. albicans* and *C. auris*, proton-detection techniques were employed, which provide significantly higher sensitivity than $^{13}$C-detection experiments extensively used for cell wall characterization[58,59,70–72]. These species exhibited highly similar NMR signatures, with β-1,3-glucan dominating the rigid fraction when detected through dipolar-coupling-based hCH spectra (Fig. 2a), which is consistent with $^{13}$C-detection results (Fig. 1d). The mobile fraction

revealed a variety of molecules with highly overlapping signals, except for a few additional galactose and glucose signals in *C. auris*, which were more abundant than those in *C. albicans* (Fig. 2b).

Most carbohydrate resonances demonstrated a high degree of degeneracy due to the polymorphic nature of carbohydrates in the cellular context, excluding β-1,6-glucan (Fig. 2c). Extending to a 3D hCCH TOCSY experiment with DIPSI-3 mixing[73] allowed the resolution of all carbon sites (Supplementary Table 2), revealing five distinct 1,2-linked mannose forms (Fig. 2d), three glucopyranose forms from β-1,3-glucan (Fig. 2e) in the cell walls, five forms of galactose or glucose derivatives, such as glucose-phosphates (Fig. 2e)[74], alongside α- and β-glucoses in the media (Fig. 2g).

The three forms of glucopyranose units in β-1,3-glucans were observed in both rigid and mobile fractions (Supplementary Tables 1 and 2) and originated from distinct sources: Type-a is the predominant form derived from the β-1,3-glucan matrix, exhibiting the highest intensity; Type-b is directly bound to chitin, as evidenced by its strong intermolecular interactions with chitin, which will be discussed later; the origin of Type-c remains unidentified. The reason for observing five types of α-1,2-linked mannose residues is unclear. However, given the diverse structural roles that α-1,2-linked mannoses need to play in the formation of mannan fibrils, the peak multiplicity observed here was likely a result of the various sugar units these residues are connected to. These α-1,2-mannoses can be connected to the α-1,6-mannan backbone, as well as to β-1,2-mannose, α-1,3-mannose, and phosphate groups found in the sidechains[27,35].

### Caspofungin treatment remodels polysaccharide composition in the *Candida* cell walls

To investigate caspofungin-induced effect on cell wall architecture, we cultured the *C. albicans* strain SC5314 in the presence of 0.015 μg/mL caspofungin and two *C. auris* strains from different genetic clades in the presence of higher drug concentrations. The *C. auris* strains included a clade-IV azole-resistant strain AR386 from Venezuela (grown in 0.05 μg/mL caspofungin)[75], and a clade-I echinocandin-resistant strain I.3 from India (grown in 0.25 μg/mL caspofungin)[76]. All drug concentrations were maintained below the respective minimum inhibitory concentrations to allow structural changes to be monitored in living cells. The use of 0.015 μg/mL caspofungin allowed *C. albicans* SC5314 to maintain ~90% of growth, providing sufficient material for NMR analysis (Fig. 3a). Complete growth inhibition for the *C. auris* AR386 strain occurred only above 32 μg/mL caspofungin, with a similar 90% of growth retained at the 0.05 μg/mL concentration used in this study. Despite exposure to a five-fold higher caspofungin concentration than AR386, the resistant *C. auris* I.3 strain sustained a similarly high level of growth. Caspofungin increased the cell wall thickness of *C. albicans* from 140 ± 11 nm to 185 ± 16 nm (Fig. 3b and Supplementary Table 3). In contrast, *C. auris* strains only showed minor changes in cell wall thickness: from 158 ± 11 nm to 162 ± 13 nm for AR386 and from 70 ± 17 nm to 75 ± 17 nm for I.3. Therefore, caspofungin-induced changes in wall dimensions were observed only in *C. albicans*.

At the molecular level, both the *C. albicans* SC5314 strain and the *C. auris* AR386 strain exhibited a notable reduction in β-1,3-glucan content following caspofungin treatment, as indicated by the decreased intensity of characteristic peaks, such as β-1,3-glucan carbon 3 (B3) at 86 ppm and carbon 4 (B4) at 68 ppm (Fig. 3c). Conversely, the chitin content increased, with higher intensities observed for the signature Ch4 and Ch2 peaks, reflecting the induction of the classical chitin upregulation compensatory mechanism that has been identified in different fungal species in response to echinocandin treatment[28,77–79]. The spectral patterns of treated *C. albicans* SC5314 and *C. auris* AR386 were nearly identical post echinocandin treatment, indicating similar

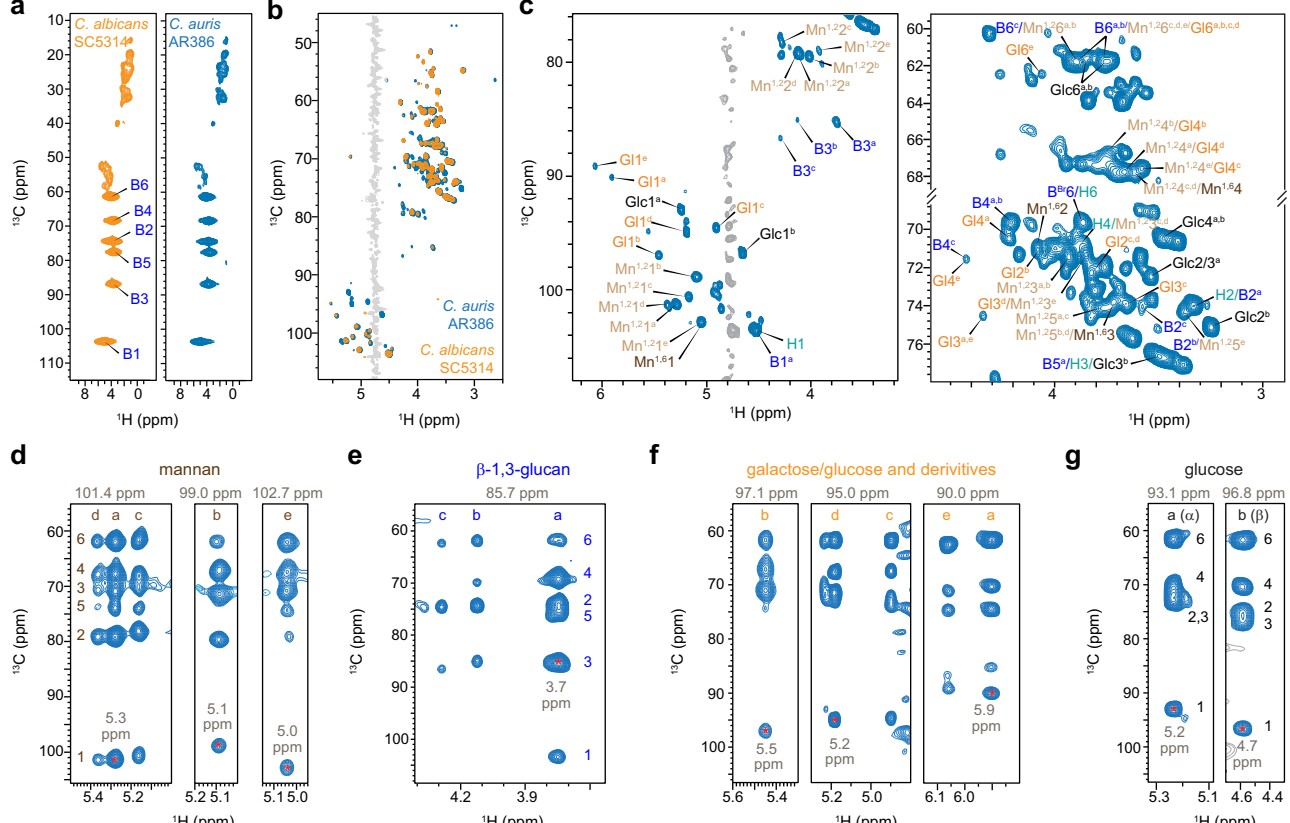

**Fig. 2 | ¹H-detected solid-state NMR resolves the polymorphic structure of cell wall carbohydrates. a** Rigid molecules of *C. albicans* (yellow) and *C. auris* (blue) cell walls detected through 2D hCH spectra. **b** Mobile molecules detected using a 2D hcCH TOCSY (DIPSI-3) experiment. Water signals are displayed in gray. **c** Selected carbohydrate regions of 2D hcCH TOCSY (DIPSI-3) spectrum of *C. auris*, showing signals from mannan (Mn), glucan (B and H), glucose (Glc), and galactose or glucose derivatives (Gl). Spectral strips (F1-F3) of key carbohydrates were extracted from 3D hCCH TOCSY (DIPSI-3) for (**d**) mannan, (**e**) β-1,3-glucan, (**f**) galactose or glucose derivatives, and (**g**) glucoses. Each carbohydrate is annotated

with letter codes (a-e), representing the number of resolvable structural forms (e.g., 5 forms for mannose and galactose; 3 forms for β−1,3-glucan). Numbers 1–6 indicate specific carbon sites within each molecule. For each strip, the ¹³C chemical shift of the F2 dimension where the F1-F3 strip was extracted from the 3D spectrum is indicated at the top, and the corresponding ¹H chemical shift is labeled, with the signal highlighted by an asterisk. The hcCH spectra were acquired on 800 MHz spectrometer under 13.5 kHz MAS, while the hCH spectra were measured on a 600 MHz spectrometer at 60 kHz MAS.

molecular structures of their rigid cell wall cores after treatment, despite their differences in wall thickness.

The 2D ¹³C-¹³C correlation spectra revealed a significant increase in the chitin content in caspofungin-treated *C. albicans* SC5314 and *C. auris* AR386, as well as a notable rise in the signals for β−1,6-glucan and mannan (Fig. 3d). These findings suggested that some mannans and β-1,6-glucans transitioned from the mobile fraction to the rigid fraction as a result of caspofungin-induced reorganization of the cell wall. The distinct chemical shift at 79 ppm is characteristic of the C2 carbon in α-1,2-linked mannose residues, which constitute the side chains of mannan fibrils in *Candida* species. Analysis of peak volumes indicated that these rigidified mannan sidechains comprised 6-7% of the rigid molecules in caspofungin-treated *Candida* species (Fig. 3e and Supplementary Table 4). β-1,6-glucan, which comprised only 1% of the rigid fraction in untreated samples, increased to 3–5% following caspofungin exposure. Simultaneously, caspofungin treatment reduced β−1,3-glucan content from 94% to 47% in *C. albicans* and from 80% to 41% in *C. auris* AR386 within the rigid portion of the cell wall. In contrast, chitin contents increased to 44-48% in both strains, consistent with the comparable 1D spectral pattern of these two samples post-treatment.

The observed rigidification of mannan and β-1,6-glucan is understandable. A recent study using high molecular weight

lectin to stain *C. albicans* cell walls—which can access only the outer layer and not penetrate the inner layer—revealed an increased level of surface-exposed chitin following caspofungin treatment[29]. This finding suggested that mannan fibrils likely interact with these surface-exposed chitin domains, thereby becoming partially rigidified through such interactions. It is important to note that only certain α-1,2-linked sidechains of mannan, rather than the more embedded α-1,6-linked backbones, interact with chitin microfibrils and undergo rigidification. Concurrently, with the depletion of approximately half of the β−1,3-glucan, more β-1,6-glucan located in the inner cell wall would likely be appropriately repositioned to physically colocalize and interact with chitin microfibrils, contributing to their increased rigidity.

None of these changes were observed in the *C. auris* I.3 strain, which exhibits echinocandin resistance due to the S639F mutation in β-1,3-glucan synthase FKS1[76]. The different levels of echinocandin susceptibility in different strains may influence how they respond to caspofungin challenge. Following caspofungin treatment, the β-1,3-glucan content in the I.3 strain remained significantly higher than that of chitin, as shown by the higher intensity of the B3 peak compared to Ch4 (Fig. 3c). Capspofungin exposure did not induce any apparent increase in the intensities of chitin, mannan, or β-1,6-glucan in 2D spectra

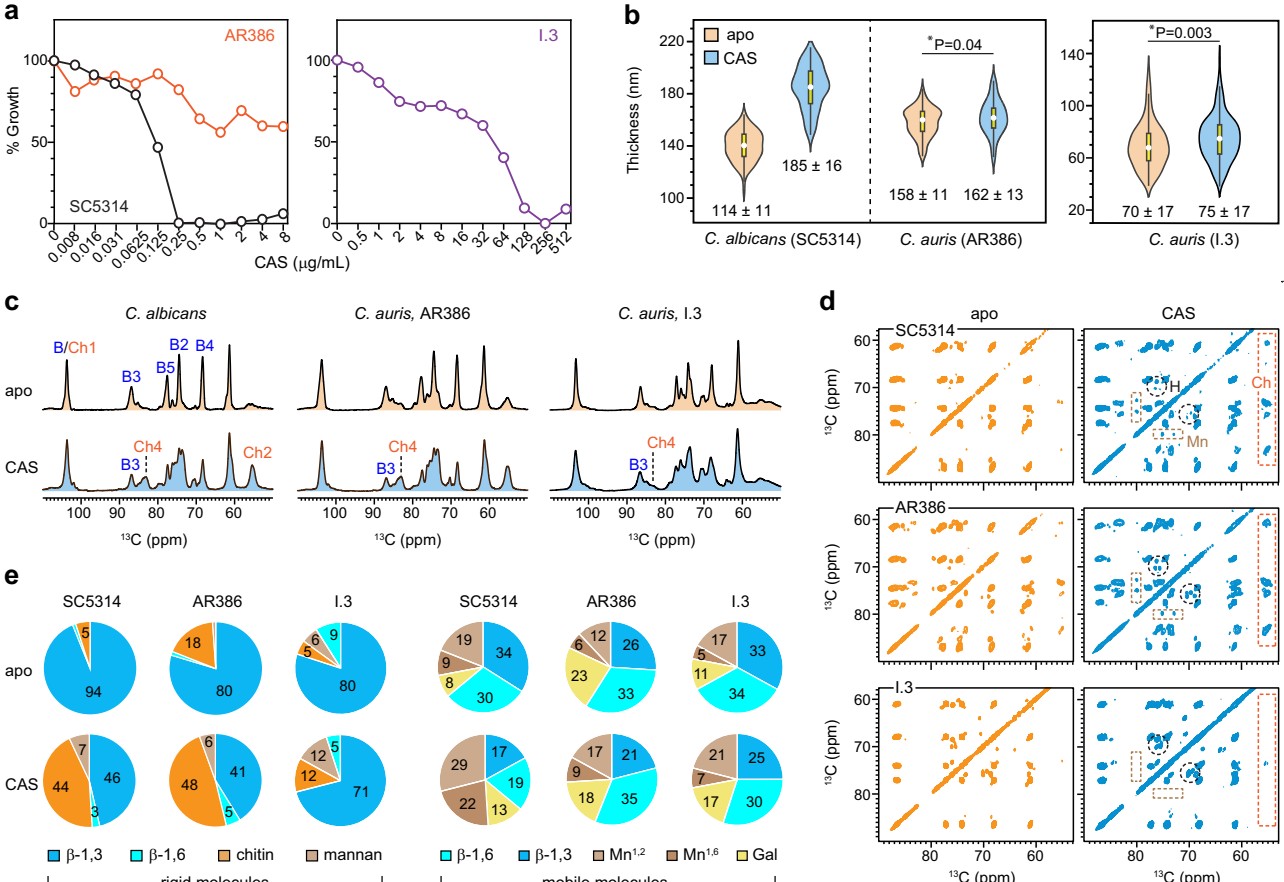

**Fig. 3 | Impact of caspofungin on *C. albicans* and *C. auris*. a** Growth profile of *C. albicans* SC5314 (black), *C. auris* AR386 (orange), and *C. auris* I.3 (purple) in response to varying concentrations of caspofungin. Percentages represent the comparison of $OD_{600}$ values between cultures grown after 24 h with and without caspofungin. **b** Cell wall thickness of *C. albicans* SC5314 and two *C. auris* strains. For each strain the untreated (apo; pale orange) and caspofungin-treated (CAS; blue) are compared. The mean ± s.d are provided. $n = 100$ for each sample. Black rectangles denote the interquartile range (IQR) from 25th to 75th percentile. White circles represent the mean of the dataset, and black vertical lines correspond to 1.5IQR. Statistical analysis was performed using a *t*-test with one tail comparison between apo and drug-treated samples for each strain. Statistically significant:

*p*-value ≤ 0.05. **c** 1D $^{13}$C cross polarization (CP) spectra of apo (top; orange) and caspofungin-treated (bottom; blue) samples of three *Candida* strains. **d** 2D $^{13}$C-$^{13}$C 53 ms CORD spectra of *C. albicans* SC5314 (top), *C. auris* AR386 (middle), and *C. auris* I.3 (bottom), with and without exposure to caspofungin. Characteristic signals of chitin (Ch), β−1,6-glucan (H), and mannan (Mn) are highlighted within dashed line boxes in orange, black, and light brown, respectively. **e** Molar composition of carbohydrate components in the rigid and mobile fractions within *Candida* cell walls. The percentage values were estimated from the volumes of resolved cross-peaks in 2D CORD spectra for the rigid fraction and the volumes of resolved signals in 2D DP J-INADEQUATE spectra for the mobile molecules. Source data are provided as a Source Data file.

(Fig. 3d). Due to its resistance, the β-1,3-glucan content in the I.3 strain remained largely unaffected (Fig. 3e and Supplementary Fig. 2), even though the drug concentration was five times higher than that applied to *C. auris* AR386 and 17 times higher than that used for *C. albicans*.

In addition to the inhibition of the biosynthesis of the primary target, β-1,3-glucan, a reduction in β-1,6-glucan level has also been reported in the literature for *C. albicans* during caspofungin treatment[80], along with an increase in the molecular size of β-1,6-glucan and cross-linking of glucan and chitin[36]. Here we also observed a 50% reduction in β-1,6-glucan content within the mobile fraction, where most of these molecules resided (Fig. 3e and Supplementary Table 5). This loss of β-1,6-glucan was compensated by an increase in the mannan content that preserved the mobile domains of the cell wall. In contrast, both *C. auris* AR386 and I.3 strains maintained similar β-1,6-glucan contents in their mobile fractions after caspofungin treatment. Thus, β-1,6-glucan, which plays a crucial role in morphogenesis, cell wall integrity, and virulence[36,81,82], appears to be a key differentiating factor in the responses of *C. albicans* and *C. auris* to caspofungin

treatment. We therefore constructed *C. auris* mutants to investigate the role of this molecule.

### *C. albicans* and *C. auris* differ in polymer dynamics and hydration response to caspofungin

Despite *C. albicans* SC5314 and *C. auris* AR386 showing nearly identical spectra for their rigid polysaccharides (Fig. 3c, e), their polymer dynamics and water retention respond differently to caspofungin treatment. The water-edited intensities ($S/S_0$) of β-1,3-glucan remained almost unchanged in *C. albicans* (Fig. 4a), indicating that the wall remained well hydrated. However, the hydration level of β-1,3-glucan was significantly decreased in *C. auris* as the $S/S_0$ dropped from 0.57 to 0.23. Chitin microfibrils in both species experienced substantial dehydration by caspofungin, with $S/S_0$ decreased from 0.53 to 0.27 in *C. albicans* and from 0.27 to 0.08 in *C. auris* (Supplementary Fig. 3 and Supplementary Table 6). This pronounced dehydration is likely driven by the formation of large crystalline chitin domains or increased proximity of poorly hydrated chitin microfibrils following the removal of nearly half of the well-hydrated β-1,3-glucan. In caspofungin-treated

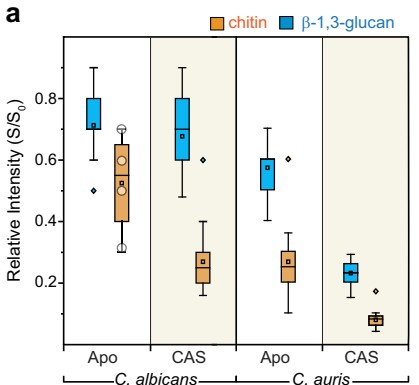
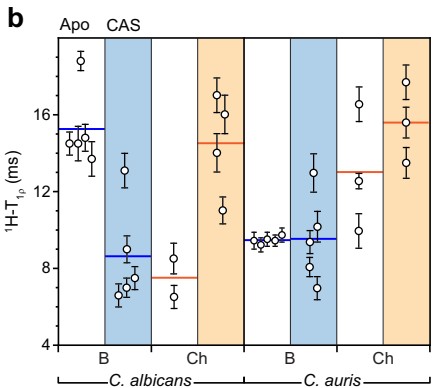
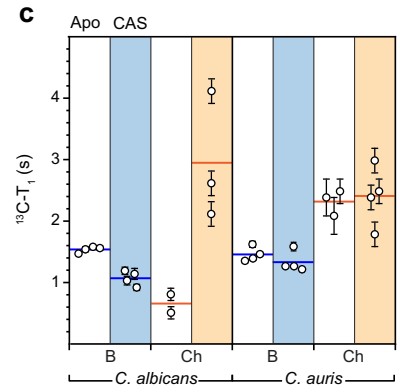

**Fig. 4 | Effect of caspofungin on polysaccharide dynamics and water retention.**
**a** Box-and-whisker plots showing the water-edited intensities ($S/S_0$) of various carbon sites in untreated and caspofungin-treated *C. albicans* (SC5314) and *C. auris* (AR386). The $S/S_0$ ratios reflect water association with β-1,3-glucan (blue; $n = 23$–25 for both *C. albicans* and *C. auris*) and chitin (orange; $n = 4$; *C. albicans* apo, $n = 10$; *C. auris* apo, $n = 14$, 15 for *C. albicans* and *C. auris* caspofungin treatment). The boxes represent the interquartile range (IQR), with whiskers extending to 1.5 times the IQR, and outliers marked by open diamonds. Mean values are represented by open rectangles, and medians by horizontal lines. The median for β-1,3-glucan data in apo *C. albicans* aligns with the lower boundary of the IQR (0.7), while in apo *C. auris*, it coincides with the upper boundary (0.6). *C. albicans* apo chitin data points are represented by open circles. **b** $^1$H-$T_{1\rho}$ relaxation time constants for β-1,3-glucan (B) and chitin (Ch) in *C. albicans* and *C. auris*, both with and without caspofungin treatment. Shaded areas correspond to datasets of treated samples: blue for *C. albicans* and orange for *C. auris*. **c** $^{13}$C-$T_1$ relaxation time constants for carbon sites within β-1,3-glucan and chitin. For both (**b**) and (**c**), error bars represent the s.d of the fit parameters and horizontal lines indicate the mean of relaxation time for each polysaccharide. Source data are provided as a Source Data file.

*C. albicans*, the remaining β-1,3-glucans were only loosely associated with chitin, preserving their hydration, while in *C. auris*, they became tightly integrated into chitin domains, leading to reduced hydration.

In apo *C. albicans*, β-1,3-glucans were initially insensitive to microsecond timescale motions, indicated by long $^1$H-$T_{1\rho}$ relaxation times with an average of 15.3 ms (Fig. 4b), which decreased to 8.6 ms following caspofungin treatment, indicating increased dynamics for β-1,3-glucan matrix (Supplementary Fig. 4 and Supplementary Table 7). Conversely, chitin in *C. albicans* became less dynamic after drug treatment, on both microsecond and nanosecond timescales, as reflected by a 2-fold increase in $^1$H-$T_{1\rho}$ and a 4-5-fold increase in $^{13}$C-$T_1$ relaxation times, respectively. Interestingly, *C. auris* showed almost no change in the dynamics of both β-1,3-glucans and chitin before and after antifungal treatment (Fig. 4b, c). The only notable difference observed for *C. auris* was that chitin became less dynamic on the microsecond timescale with a moderate increase of $^1$H-$T_{1\rho}$ from 13.0 ms to 15.6 ms.

### Chitin-associated β-glucans diverge from the traditional triple-helix model structure

To investigate whether caspofungin treatment affects the spatial organization of polysaccharides within the mechanical core of *C. auris* cell wall, we introduced the biradical AsymPolPOK[83] into intact cells, achieving a 57-fold enhancement in the NMR sensitivity for carbohydrates (Fig. 5a), which reduced measurement time by 3,249-fold. AsymPolPOK and other biradicals are small (2–3 nm), water-soluble molecules that diffuse through porous cell walls without disrupting their structure, enabling widespread use in the structural analysis of plant, fungal, algal, and bacterial cell wall materials as well as intact mammalian cells[84–90]. Sensitivity enhancement is achieved by transferring electron spin polarization to $^1$H nuclei in the biradical, followed by $^1$H spin diffusion across the solvent and finally to biomolecules, resulting in homogeneous hyperpolarization over distances of at least tens of nanometers[84]. Proteins and lipids exhibited a lower, 30-fold enhancement because, although some may contribute to cell wall structuring, the majority are predominantly located in the plasma membrane or intracellularly, making them less exposed than cell wall polysaccharides. Under DNP conditions at cryogenic temperatures, signals from disordered components were broadened out, leaving only signals from partially ordered molecules detectable[91]. These included three major components: β-1,3-glucan, chitin, and unexpectedly, mannan. Notably, the mannan signals predominantly originated from α-1,2-linked sidechains, identified by unique carbon 1 and carbon 2 signals at 101-102 ppm and 81-82 ppm, respectively, and later confirmed via INADE-QUATE and PAR spectra (Supplementary Fig. 5a). This suggests that some chitin, β-1,3-glucan, and mannan sidechains are physically packed to maintain a partially ordered core structure.

Despite the short 5 ms PAR recoupling period, intermolecular cross-peaks were observed between highly populated interfaces between distinct polysaccharides (Supplementary Fig. 5b, c). The most prominent interactions included those between chitin and β-1,3-glucan. Specifically, two forms of β-1,3-glucan can be differentiated based on cross peaks between carbons 4 and 3 (B4-3 and B$^b$4-3), with the $^{13}$C chemical shifts of the carbon 3 sites at 85.0-86.8 ppm and 83.5-84.5 ppm, respectively (Fig. 5b). These chemical shifts are slightly lower than those reported at room temperature (Fig. 2e) due to the cryogenic conditions used during the DNP experiment. Among these two forms, the former corresponds to type-a β-1,3-glucan observed at room-temperature (Fig. 2e), representing the bulk β-1,3-glucans in the model triple-helix configuration[67–69], exhibiting no interaction with chitin. The latter (B$^b$) corresponds to the type-b β-1,3-glucan, a minor population of β-1,3-glucans closely associated with chitin, as evidenced by the ChMe-B$^b$3 cross-peak happening between chitin methyl groups and the carbon 3 sites of these specific β-1,3-glucans. The 2-ppm upfield shift observed at the C3 site involved in the glycosidic linkage was attributed to conformational perturbations in the β-1,3-glucan triple-helical structure, probably caused by the collapse of β-1,3-glucan onto chitin microfibril surfaces. The data clearly demonstrated that β-1,3-glucan adopts distinct structures when bound to chitin microfibrils compared to when partitioned into the solvent, with the former representing only a small fraction. The unique structure of chitin-bound β-1,3-glucan was consistently observed in both apo and caspofungin-treated *C. auris* cell walls (Supplementary Figs. 5 and 6). Therefore, the tight association between chitin and β-1,3-glucan in the rigid core

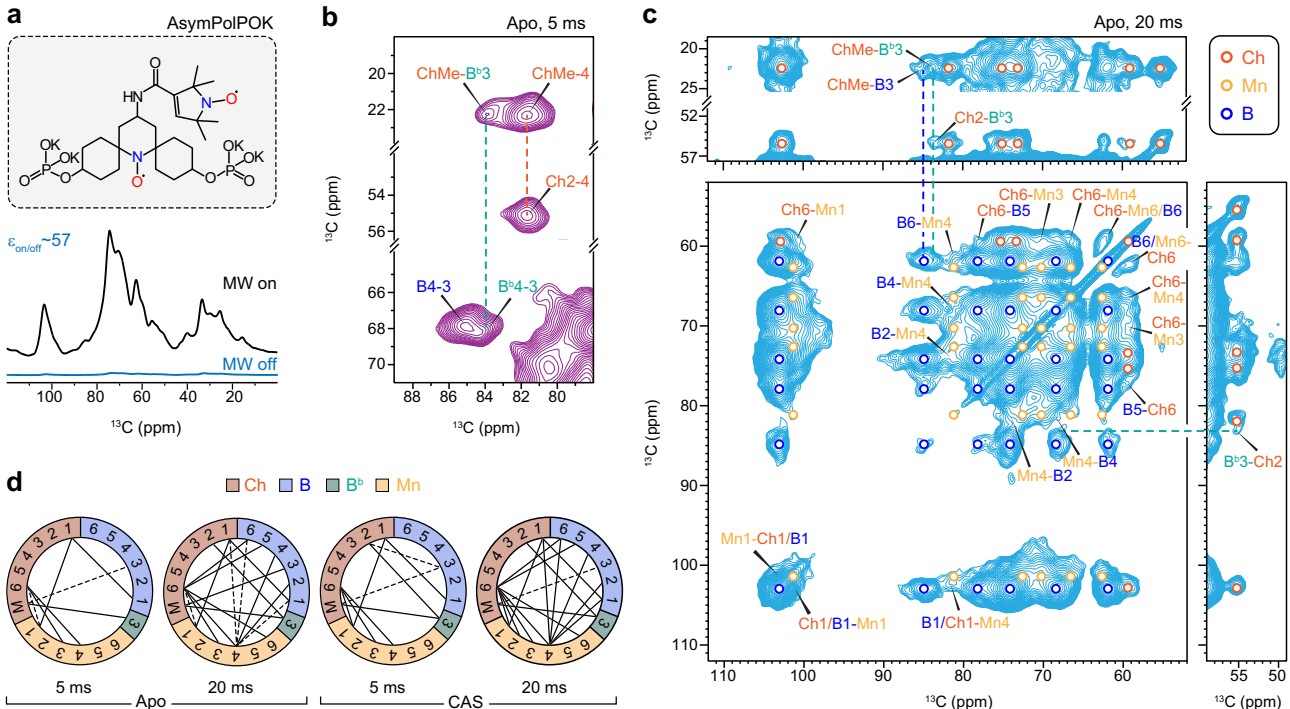

**Fig. 5 | DNP-enhanced analysis of polysaccharide organization in *C. auris*.**
**a** MAS-DNP of apo *C. auris* AR386 using the AsymPolPOK biradical (top) resulting in a 57-fold (bottom) enhancement in NMR sensitivity. The enhancement factor ($\varepsilon_{on/off}$) is calculated by comparing the intensity of spectra recorded with and without microwave (MW) irradiation. **b** DNP 2D $^{13}$C-$^{13}$C correlation spectra of apo *C. auris* AR386 with 5 ms PAR recoupling time. An intermolecular cross-peak is noted between the chitin methyl group and carbon 3 of a minor β-1,3-glucan form (B$^b$; cyan), rather than the bulk form (B; blue). **c** DNP 2D $^{13}$C-$^{13}$C correlation spectrum of apo *C. auris* AR386 measured with a longer PAR mixing of 20 ms. intramolecular

correlations within chitin, mannan, and β-1,3-glucan are marked using open circles (orange, yellow, and blue, respectively). Long-range intermolecular cross-peaks between these polysaccharides are labeled. **d** Summary of intermolecular interactions among chitin (dark orange), β-1,3-glucan (blue), and mannan (yellow). The carbon 3 site of the special β-1,3-glucan form (B$^b$) is highlighted in green. Dashed lines indicate single interactions (e.g. ChMe-B$^b$3), while solid lines show bidirectional interactions (e.g. ChMe-B$^b$3 and B$^b$3- ChMe) to account for the direction of polarization transfer. Source data are provided as a Source Data file.

represents a conserved structural feature that remains unaffected by echinocandin treatment, which is crucial for the mechanical strength of the cell wall and fungal survival.

## The rigid cell wall core of *C. auris* remains unaffected by caspofungin treatment

The minimal impact of caspofungin on the polysaccharide dynamics and hydration profiles in *C. auris* raises the question of whether any structural alterations occurred in the rigid core of the cell wall architecture. Extending the PAR duration to 20 ms led to the detection of numerous intermolecular cross-peaks (Fig. 5c). For example, interactions between the chitin methyl group and the carbon 3 sites of both the special form and bulk form of β-1,3-glucans (ChMe-B$^b$3 and ChMe-B3, respectively) were observed. However, the special form (B$^b$3) showed stronger interactions with chitin, as indicated by the higher intensity of ChMe-B$^b$3 compared to ChMe-B3, and the presence of cross-peaks for Ch2-B$^b$3 and B$^b$3-Ch2, without corresponding peaks for the bulk β-1,3-glucan. Additionally, interactions between mannan and both chitin and β-glucan were observed, such as Ch6-Mn1 and B6-Mn4. These data demonstrate that a large portion of chitin, β-1,3-glucan, and mannan sidechains are physically associated within the cell wall of *C. auris* AR386. This structural paradigm is also likely to be applicable to *C. albicans*, given the high similarity between these two strains in the apo state.

The most surprising observation was that caspofungin treatment did not disrupt the physical organization of ordered polysaccharides within the *C. auris* cell wall. This is evidenced by the highly comparable polymer contact map, which summarizes 98 intermolecular cross-

peaks observed in both apo and caspofungin-treated samples (Fig. 5d), despite the caspofungin-treated samples showing a few additional interactions (Supplementary Fig. 6). This finding aligned with the observed similarity in polymer dynamics and hydration profiles between the apo and caspofungin-treated *C. auris*, collectively indicating that the structure of the rigid core remained unperturbed, even though the molecular composition of the cell wall has been reorganized.

## β-1,6-glucans modulate the susceptibility of *C. auris* to echinocandins

The next step involved evaluating the association between β-1,6-glucan and the activity of echinocandins, given that this molecule was implicated in the observed differences between *C. albicans* and *C. auris* (Fig. 3e). The biosynthesis and structure of β-1,6-glucans is also a crucial but under-investigated component of the *Candida* cell wall, which have been shown recently to be highly responsive to environmental factors such as pH, salt, and the presence of antifungals, and can be regulated to compensate for defects in *O*-/*N*-mannan elongation[36]. We identified in the *C. auris* B8441 genome two contiguous genes with 57% homology to the β-1,6-glucan synthase gene *KRE6* of *C. albicans*[82]. These genes were hereafter referred to as *KRE6a* (*B9J08_002262*) and *KRE6b* (*B9J08_002263*). Attempts to generate a double *KRE6a* and *KRE6b* deletion strain in *C. auris* AR386 were unsuccessful (Supplementary Fig. 7 and Supplementary Table 8); although the deletion cassette integrated into the *KRE6a/b* loci, both genes persisted, suggesting that these genes are essential in *C. auris*. Similarly, *KRE6* has been considered essential in *C. albicans*, but the recent success in

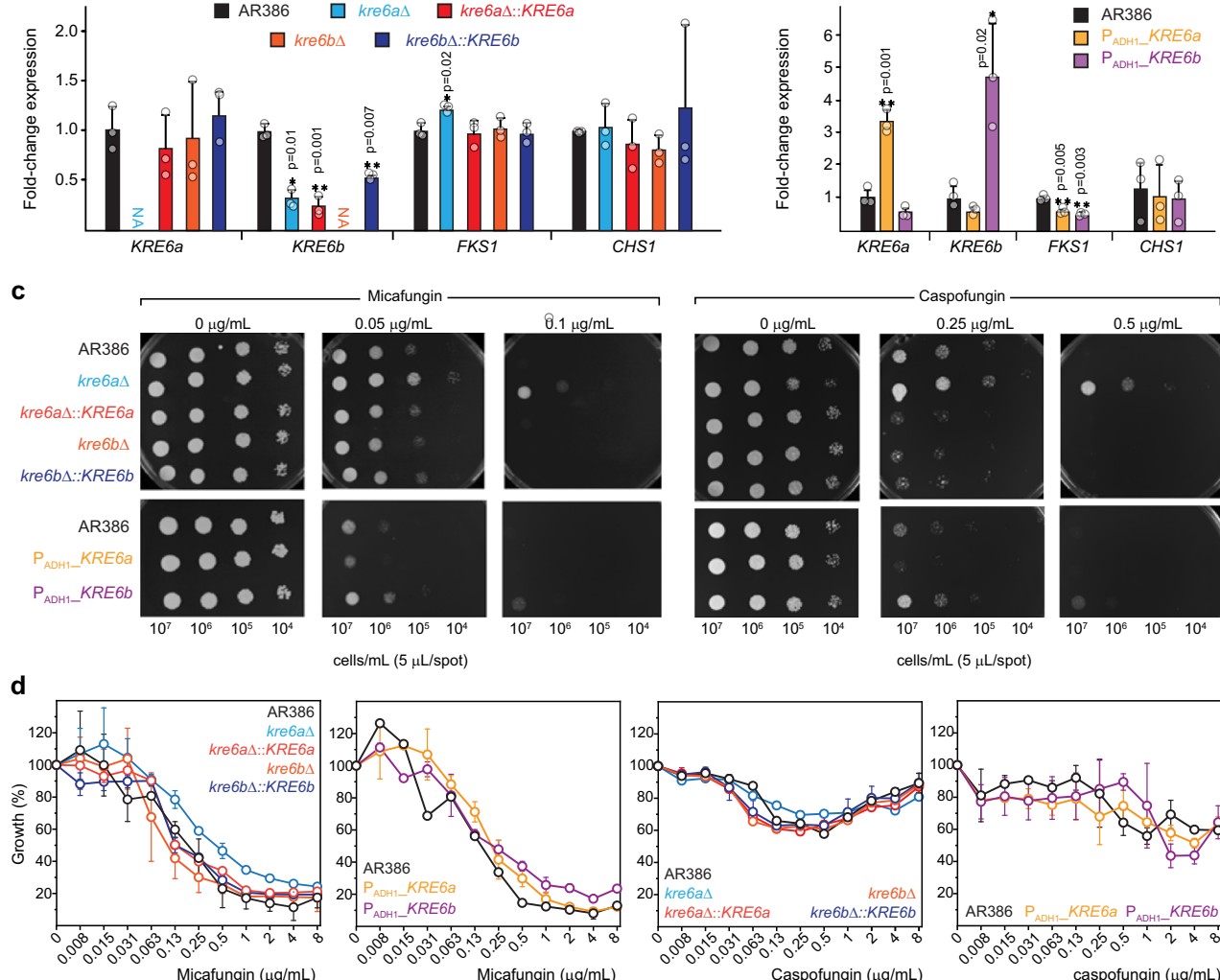

**Fig. 6 | Susceptibility to micafungin and caspofungin in *KRE6* mutant strains of *C. auris*.** Relative expression of genes involved in cell wall biosynthesis (*KRE6a*, *KRE6b*, *FKS1* and *CHS1*) by real-time reverse transcription RT-PCR are compared for (**a**) *KRE6a* and *KRE6b* deletion and complemented strains, as well as (**b**) *KRE6a* and *KRE6b* overexpressing strains. Black: AR386; cyan: *kre6aΔ*; orange: *kre6bΔ*; red: *kre6aΔ::KRE6a*; dark blue: *kre6bΔ::KRE6b*; yellow: P_ADHL_*KRE6a*; purple: P_ADHL_*KRE6b*. Results are expressed as fold-change compared to the wild-type AR386. Bars represent the mean with s.d. of three biological replicates. Data points are represented by open circles. Statistical analysis was performed using a *t*-test with two tails compared to AR386. Statistically significant: **p*-value ≤ 0.05. ***p*-value ≤ 0.01. **c** Susceptibility to micafungin (left) and caspofungin (right) was performed by spotting assays in all the *KRE6a/b* mutants. Spotting assays were performed with serial dilutions of yeast cells spotted on YEPD agar plates containing different concentration of micafungin or caspofungin. Plates were incubated for 24 h at 37 °C. **d** Susceptibility to micafungin and caspofungin performed using microbroth dilution assay (EUCAST protocol) in all the KRE6a/b mutants. Growth is expressed as the percentage of OD_600 compared to that of the wild-type AR386 in the absence of drug. Open circle represent the mean of biological duplicates and error bars are s.d. Source data are provided as a Source Data file.

creating a homozygous *KRE6* deletion mutant under specific conditions has raised questions about the essentiality of this gene[82].

We generated two single-deletion strains (*kre6aΔ* and *kre6bΔ*) targeting *KRE6a* and *KRE6b*, respectively, as well as their complemented strains (Supplementary Figs. 8–12). We measured the expression of *KRE6a*, *KRE6b*, *FKS1* (β-1,3-glucan synthase gene, *B9J08_000964*) and *CHS1* (chitin synthase gene, *B9J08_003879*) in these deletion and complementation by real-time reverse transcription PCR (RT-PCR) (Fig. 6a). This experiment confirmed the loss of *KRE6a* and *KRE6b* expression in *kre6aΔ* and *kre6bΔ*, respectively. Complementation of *KRE6a* (*kre6aΔ::KRE6a*) resulted in full restoration of *KRE6a* expression, while complementation of *KRE6b* (*kre6bΔ::KRE6b*) achieved partial restoration of *KRE6b* expression. This could be due to some loss of the distal portion of the *KRE6b* promoter resulting from the insertion of the complementation cassette. Because the *KRE6a* and *KRE6b* loci are separated by only 764 bp, the

complementation cassette was inserted at -426 bp from the start codon of *KRE6b* (Supplementary Fig. 12).

Interestingly, we also observed a significant downregulation (0.26-fold) of *KRE6b* in *kre6aΔ*, which suggested an auto-regulation loop of *KRE6a* over *KRE6b* (Fig. 6a). This effect persisted in the *kre6aΔ::KRE6a* strain supporting the hypothesis that the entire 764-bp intergenic region between the *KRE6a* and *KRE6b* loci is essential to maintain proper *KRE6b* expression. A slight but significant increase of *FKS1* expression was observed in *kre6aΔ*, but not in *kre6bΔ*. This result suggests a compensatory effect resulting from the downregulation of all β-1,6-glucan synthase genes (*KRE6a* and *KRE6b*) in *kre6aΔ*, which was not the case in *kre6bΔ* (maintaining normal level of *KRE6a* expression). Complementation of *KRE6a* could reduce *FKS1* expression to its basal level. Expression of *CHS1* was not affected in the deletion strains and their respective complemented strains.

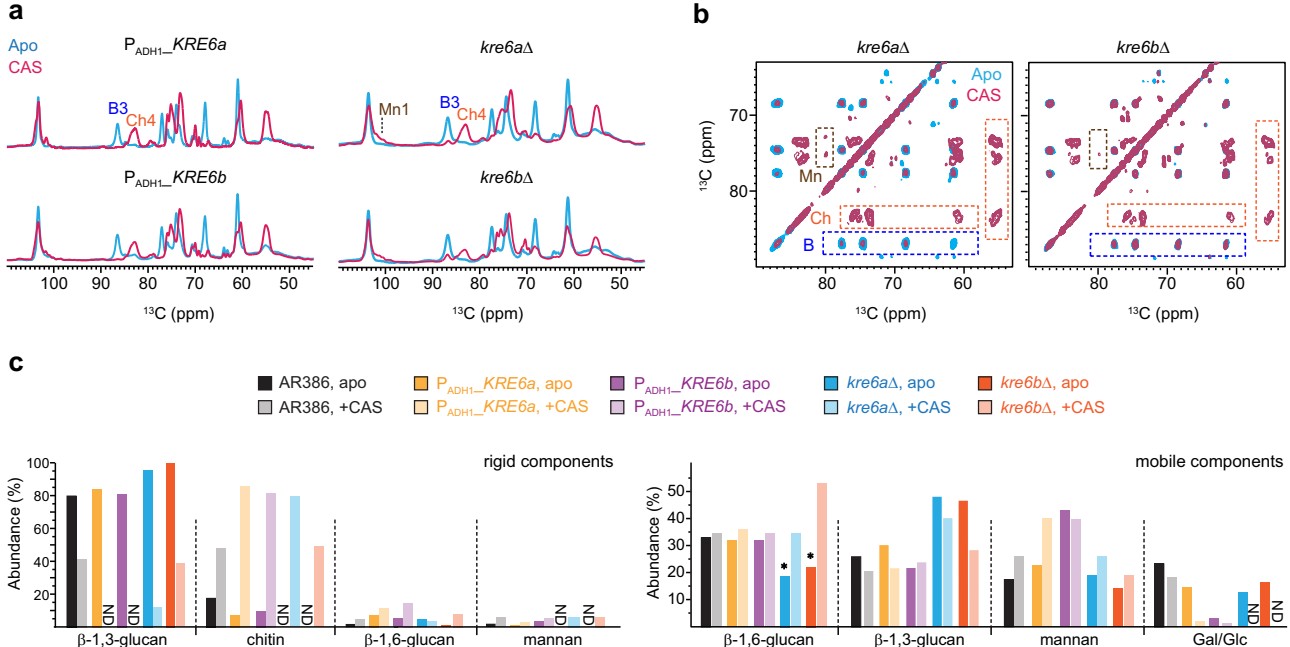

**Fig. 7 | Modulated cell wall composition in *KRE6* mutant strains of *C. auris*. a** 1D $^{13}$C CP spectra of the four mutant samples prepared without (blue) and with (magenta) 0.5 µg/mL caspofungin. **b** 2D $^{13}$C CORD spectra of *kre6a*Δ (left) and *kre6b*Δ (right) samples prepared without (blue) and with (magenta) 0.5 µg/mL caspofungin. Signature spectral regions are highlighted by dashed line boxes for chitin (Ch; orange), β-1,3-glucan (B; blue), and mannan (Mn; brown). **c** Molar composition of rigid (left) and mobile (right) carbohydrates estimated from peak volumes in 2D CORD and DP J-INADEQUATE spectra, respectively. Black: AR386; yellow: P$_{ADH1}$_*KRE6a*; purple: P$_{ADH1}$_*KRE6b*; blue: *kre6a*Δ; orange: *kre6b*Δ. The corresponding pale color is used for caspofungin-treated sample of each strain. For each carbohydrate, data is shown for tens samples: the AR386 strain and four mutants prepared with and without caspofungin. Asterisks highlight the decreased content of β–1,6-glucan in *KRE6* deletion mutants, which are recovered by caspofungin treatment. Source data are provided as a Source Data file.

Next, we generated two strains (P$_{ADH1}$_*KRE6a* and P$_{ADH1}$_*KRE6b*), in which the *KRE6a* and *KRE6b* genes were placed under the control of the *ADH1* promoter for their overexpression (Supplementary Fig. 13). RT-PCR analyses confirmed the overexpression of *KRE6a* in P$_{ADH1}$_*KRE6a* and *KRE6b* in P$_{ADH1}$_*KRE6b* (3.3-fold and 4.8-fold, compared to AR386, respectively) (Fig. 6b). However, *KRE6a* overexpression did not alter *KRE6b* expression and, inversely, there was no impact of *KRE6b* overexpression on *KRE6a* expression. *FKS1* expression was significantly downregulated in both P$_{ADH1}$_*KRE6a* and P$_{ADH1}$_*KRE6b*, which suggests regulatory mechanisms between β-1,6-glucan and β-1,3-glucan synthesis pathways. No significant change of *CHS1* expression was observed in any of the overexpressing strains.

To link these RT-PCR results with a phenotypic impact on echinocandin susceptibility, we tested the susceptibility to micafungin and caspofungin of all these mutant strains (Fig. 6c, d). In spotting assays, we observed a decreased susceptibility to micafungin and caspofungin of *kre6a*Δ, but not of *kre6b*Δ and the complemented strains, compared to their background strain AR386 (Fig. 6c). These results were confirmed in microbroth dilution assay with micafungin MIC$_{50}$ of 0.5 and 0.25 µg/mL for *kre6a*Δ and AR386, respectively (Fig. 6d). Results were difficult to interpret for caspofungin because of a paradoxical growth effect in all strains with none of them achieving 50% inhibition. However, the maximal degree of growth inhibition was decreased in *kre6a*Δ compared to AR386 (29.7% vs 42.2% at 0.5 µg/mL of caspofungin). Inversely, we observed a very slight increase of susceptibility to micafungin and caspofungin of P$_{ADH1}$_*KRE6a* in spotting assay, which was not confirmed in broth microdilution assay. In conclusion, only the *kre6a*Δ strain was considered to exhibit a significant change in echinocandin susceptibility (i.e. decreased susceptibility). These results are consistent with the RT-PCR analyses showing *FKS1* overexpression in *kre6a*Δ.

Although this increase of *FKS1* expression was modest (1.2-fold), it could be sufficient to induce the slight decrease of echinocandin susceptibility of this strain. However, we did not observe significant changes of echinocandin susceptibility in the overexpressing strains (P$_{ADH1}$_*KRE6a* and P$_{ADH1}$_*KRE6b*) despite some decrease of *FKS1* expression. Taken together, these results confirm the compensatory effects between the β-1,6-glucan and β-1,3-glucan synthesis pathways.

Solid-state NMR spectra of all four mutant strains treated with 0.5 µg/mL caspofungin revealed significant perturbation in their rigid polysaccharides. The β-1,3-glucan content was reduced, as indicated by the absence of carbon 3 signals (B3) in the 1D $^{13}$C CP spectra (Fig. 7a), while chitin synthesis was upregulated, evidenced by an increased carbon 4 peak (Ch4). However, in contrast to the two *KRE6* overexpression strains, where β-1,3-glucans were entirely depleted from the rigid fraction by caspofungin, both *kre6a*Δ and *kre6b*Δ strains retained a portion of rigid β-1,3-glucans after treatment (Fig. 7a, b). Intensity analysis revealed that the residual β-1,3-glucans constituted ~10% and 35% of the rigid cell wall polymers in the *kre6a*Δ and *kre6b*Δ strains, respectively (Fig. 7c, left). Additionally, the emergence of mannan within the rigid fraction of the cell wall was observed in both *kre6a*Δ and *kre6b*Δ strains following caspofungin treatment (Fig. 7a, b). These spectral changes highlighted the compensatory responses to caspofungin treatment, characterized by a reduction in β-1,3-glucan levels, an increase in chitin content, and enhanced interactions between chitin microfibrils and mannan side chains. These findings aligned with previous observations in *Candida* and *Aspergillus* species[50,77,78] and our results described earlier in this report for wild-type *C. albicans* and *C. auris* strains (Fig. 3d).

Compared to the wild-type strain, we observed a significant decline of β-1,6-glucans in the mobile fraction (Fig. 7c, right), where most of this molecule resides, for both *kre6a*Δ and *kre6b*Δ strains,

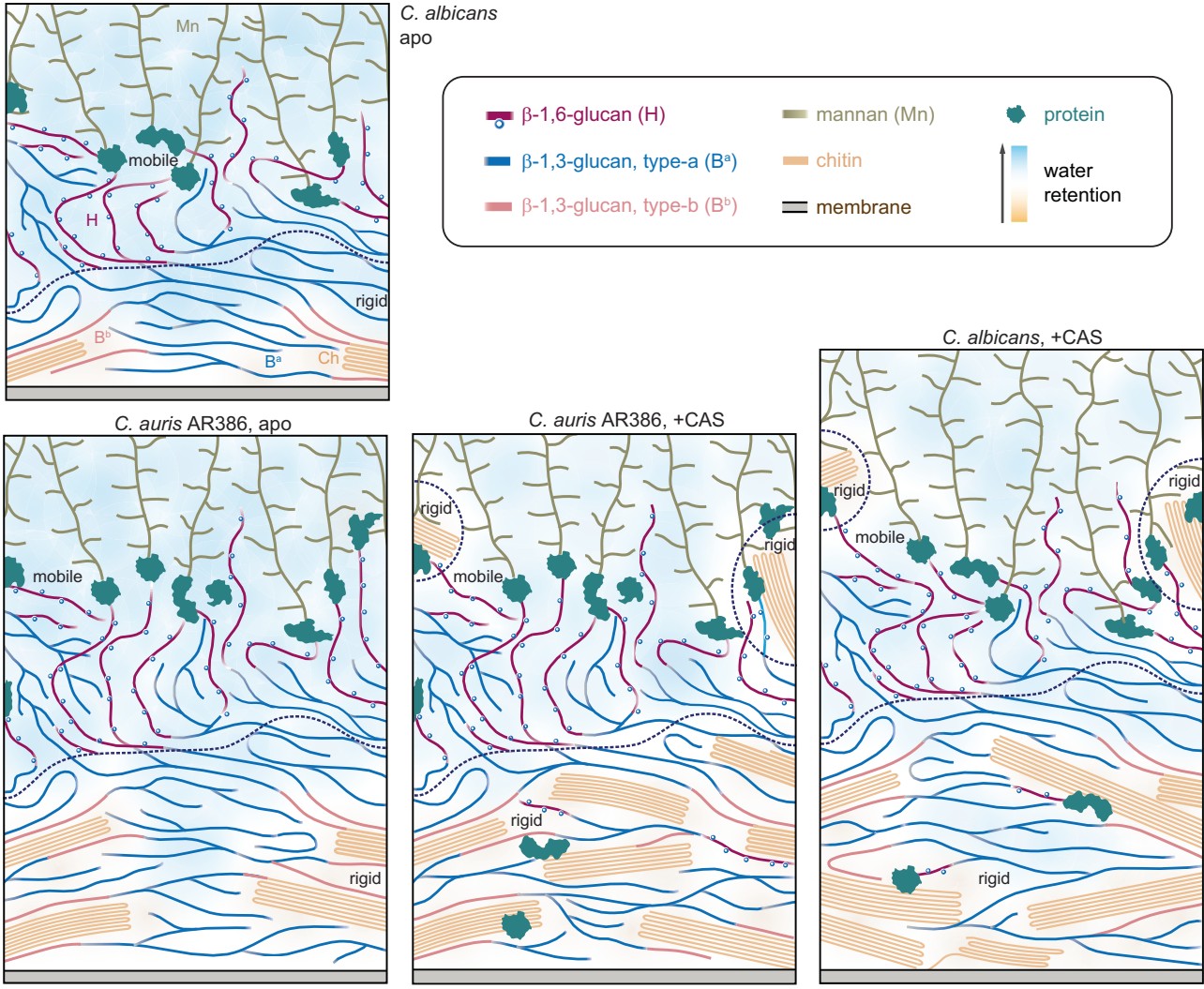

**Fig. 8 | Schematic illustration of cell wall structure and remodeling in *Candida* species.** Four panels are presented for *C. albicans* and *C. auris* AR386 in the presence and absence of caspofungin. The schematic plots are based on information on cell wall thickness from TEM measurements and molecular composition, dynamics, hydration gradient, and intermolecular interactions from solid-state NMR results from this study, as well as literature reported molecule location and linkage patterns[35,36]. The sizes and orientations of the polymers are hypothetical. Dashed lines are used to distinguish rigid and mobile domains within the cell wall. Different carbohydrates are color-coded. Blue: type-a β-1,6-glucan, pink: type-b β-1,6-gluca; dark yellow: mannan; light orange: chitin: gray: membrane; green: protein; purple: β-1,6-glucan. Open blue circles represent the β−1,3-glucoside branching along the β−1,6-glucan backbone, as recently determined[36]. The background color gradient from orange to blue highlights the water gradient from poorly hydrated to well-hydrated.

which is expected for these single-deletion mutants where the β-1,6-glucan biosynthesis is impaired. The lower β-1,6-glucans content is accompanied by an increased level of β-1,3-glucan in the same, mobile phase when compared to the wild-type strain. Exposure to caspofungin further altered the composition of the mobile matrix by increasing β-1,6-glucans, from 19% to 34% for *kre6aΔ* and from 23% to 53% for *kre6bΔ*, while reducing the content of β-1,3-glucan within each deletion mutant (Fig. 7c, right). These findings suggested that β-1,6-glucan and β-1,3-glucan have partially interchangeable roles in forming the mobile matrix, likely compensating for the loss of each other. Interestingly, the addition of caspofungin led to similar or even higher β-1,6-glucan levels in the single deletion mutants compared to AR386: the β-1,6-glucan level in the mobile fraction of caspofungin-treated cell walls was 35%, 34%, and 53% for wild-type, *kre6aΔ*, and *kre6bΔ* samples, respectively (Fig. 7c, right). Therefore, under caspofungin stress, a single KRE6 protein, either KRE6a or KRE6b (with KRE6b playing a more prominent role), is sufficient to support β-1,6-glucan biosynthesis.

## Discussion

Solid-state NMR data provides detailed biophysical insights into polymer dynamics, hydration, and molecular packing, which can be integrated with extensive biochemical and imaging results that map out the chemical composition, linkages, and spatial distribution of molecules in the cell wall (Fig. 8)[27,34,35]. In the inner layer, a rigid scaffold is formed by dehydrated chitin microfibrils and certain hydrated β-1,3-glucan, which are either covalently or physically associated (Figs. 1d and 3e). Dehydration of chitin reflects its antiparallel hydrogen-bonded packing of the linear carbohydrate chains that exclude most water molecules. Notably, β-1,3-glucan in direct contact with chitin (labeled as B^b in Fig. 8) adopts a conformation distinct from the typical triple-helix model adapted by bulk β-1,3-glucans (labeled as B^a in Fig. 8) in *Candida* cell walls (Fig. 5b). Meanwhile, a smaller fraction of β-1,3-glucan, along with most β-1,6-glucans, constitutes the mobile phase of the inner layer, contributing to its flexibility (Fig. 1e). The mannan components exhibited flexible motion (Fig. 3e), whether as long *N*-mannan fibrils extending through the outer layer or as shorter *O*-linked

mannans associated with cell wall proteins in the inner region[27]. The protein and lipid components primarily reside in the mobile phase, exhibiting dynamic behavior similar to that of mannan (Supplementary Figs. 14 and 15). While specific protein and lipid signals persist through the PAR duration, indicating a partially ordered structure, they do not show any detectable cross-peaks with carbohydrate components (Supplementary Figs. 5 and 6). This suggested that the protein-carbohydrate interface, likely involving mannoproteins, occurs only in limited regions of the overall cell wall. Some mannan fibrils rely on their sidechains to maintain spatial proximity to chitin and β-1,3-glucan, forming interactions that can be observed when the sample is subjected to cryogenic temperatures under DNP conditions (Fig. 5c). This structural arrangement applies to both *C. auris* and *C. albicans* in their apo states.

When a sub-MIC concentration of caspofungin was applied, the profile of polymer dynamics became more complex in *C. albicans* and *C. auris* AR386. Both species showed an increase in chitin content and a reduction in β-1,3-glucan (Fig. 8 and Fig. 3c), as expected for the canonical compensatory response to echinocandins. Additionally, some β-1,6-glucan and *N*-mannan sidechains became rigidified, likely due to new interactions with chitin (Fig. 3d). This may be a result of chitin relocation from the inner to the outer layer, which also enhanced chitin-protein association, as recently reported (Fig. 8)[29]. Both *C. albicans* and *C. auris* AR386 exhibit reduced water permeability (Fig. 4a), a common response previously also observed in several *Aspergillus* species under stress[50,53]. However, *C. auris* shows more pronounced dehydration in its rigid core compared to *C. albicans* (Fig. 8), leading to a more robust cell wall that provides better protection against stress. Notably, *C. albicans* fully restructured its cell wall, evidenced by a 1.5-fold increase in wall thickness, while *C. auris* strains showed no equivalent phenotypic responses (Fig. 3b). After caspofungin treatment, chitin in *C. albicans* was significantly less flexible on both nanosecond and microsecond timescales, while β-1,3-glucan became more dynamic, indicating a better separation of these two polymers (Fig. 4b, c). In contrast, *C. auris* AR386 showed no significant changes in the dynamics or intermolecular packing of these molecules (Fig. 5d). Meanwhile, an echinocandin-resistant strain of *C. auris*, I.3, exhibited no detectable changes in cell wall thickness, molecular composition, or polymer dynamics following treatment with a low concentration of caspofungin (Fig. 3b-e). This inertness likely allows the strain to maintain cell wall functions essential for stress resistance and viability.

Although both single-deletion mutants of *C. auris* exhibited a 40-50% reduction in β-1,6-glucans compared to the wild-type strain, the *kre6b*Δ mutant overproduced β-1,6-glucans in response to caspofungin, leading to a reduction in β-1,3-glucans within the soft matrix (Fig. 6g). Additionally, caspofungin-treated *kre6b*Δ showed a 40% lower chitin content compared to the treated *kre6a*Δ sample. These changes likely resulted in weaker cell walls in *kre6b*Δ compared to *kre6a*Δ, which displayed reduced susceptibility to echinocandins (Fig. 6b). These observations suggest that the ability of *kre6a*Δ to withstand echinocandin stress is a cumulative outcome of at least three key factors: (1) accumulation of a substantial amount of chitin fibrils to support rigidity, (2) slightly overexpressed *FKS1* and retention of certain β-1,3-glucan in the rigid core, likely playing a crucial role in stabilizing chitin or crosslinking molecules, and (3) maintenance of mobile matrix with a higher proportion of β-1,3-glucan relative to β-1,6-glucan. Any disruption to these structural features compromises the integrity of the cell wall and its ability to support reduced susceptibility to echinocandins.

Unexpectedly, β-1,6-glucan levels remain unchanged in both $P_{ADH1}$_KRE6a and $P_{ADH1}$_KRE6a mutants compared to the wild-type strain, even after caspofungin treatment (Fig. 6g). If the Kre6 proteins require stoichiometric levels of other ancillary proteins to form an active enzymatic complex, then overexpression may not result in a

significant increase in enzymatic activities. Alternatively, *KRE6a* and *KRE6b* may need to be present in equimolar amounts so that the overexpression of individual genes does not impact the overall biosynthesis process.

The $^1$H-detection data revealed structural polymorphism of β-1,3-glucans in native and intact cell walls (Fig. 2e), while the DNP data further provided the evidence that chitin-bound β-1,3-glucan exhibits a structural organization distinct from bulk β-1,3-glucan, deviating from the conventional triple-helix model (Fig. 5b). Such structural rearrangements have been observed in the xylan of the secondary cell walls across various plant species, where folding onto the flat surfaces of cellulose microfibrils induces a conformational shift from a three-fold helical structure to a flat, two-fold conformation[92–94]. However, this effect has not previously been documented for fungal polysaccharides. It remains unclear whether this structural variation stems from covalently linked chitin-β-1,3-glucan, which is a unique feature of the heavily crosslinked fungal cell wall[36,95], or due to the physical stacking of these polymers, or a combination of both. Future investigations should employ solid-state NMR in combination with chitin/β-1,3-glucan mutants and consider chemical shift calculations from structural models to better elucidate this relationship, as well as to validate it across different fungal species including *Aspergillus* and *Cryptococcus* species.

This high-resolution insight into *Candida* cell wall organization also raises questions that require investigations in subsequent studies. Specific structural questions arise regarding the enigmatic role of mannan backbones, which, despite surprisingly found to be mobile in intact cells, are microscopically observed to form the scaffold of mannan fibrils extending through the outer cell wall layer[27]. The functional roles of other less frequent monosaccharide units that are part of the mannan fibrils, such as α-1,3-linked and β-1,2-linked residues, must also be identified and examined. Another underexplored molecule is β-1,6-glucan[36,82], which appears to play critical roles in interacting with both mannan and chitin, as suggested by solid-state NMR results, and may exhibit different behaviors between *C. albicans* and *C. auris* in response to echinocandins. The creation of deletion mutants targeting these two polymers, followed by solid-state NMR analysis, could provide a more comprehensive understanding of *Candida* cell wall structures under stress.

Finally, the primary biological question pertains to the structural transitions of the cell wall during morphological changes. Certain *Candida* species can transition from yeast forms to filamentous structures, producing true hyphae in species like *C. albicans*, *C. dubliniensis*, and *C. tropicalis*, or *pseudohyphae* in species such as *C. auris* and *C. parapsilosis*[96–98]. The pathogenicity and immunogenicity of *Candida* species are closely linked to these morphological alterations[99]. In addition, it is necessary to examine whether other major *Candida* species, including *C. tropicalis*, *C. parapsilosis*, *C. guilliermondii*, *C. glabrata*, and *C. krusei*, adapt similar structural principles as *C. albicans* in constructing their cell walls and responding to drug-induced stress. This inquiry is particularly important given that these species rank among the leading causative agents of systemic candidiasis. Notably, *C. tropicalis* is a significant contributor to invasive candidiasis, with evidence suggesting that its mortality rates are comparable to or exceed those associated with *C. albicans*. Moreover, it is crucial to investigate the link between cell wall structure and drug resistance in *C. auris*, because the strains resistant to different antifungal classes may exhibit distinct structural characteristics—a hypothesis supported by the observed differences between strains AR386 and I.3 in this study. Determining whether a unique cell wall structure exists in the strains resistant to all three major classes of antifungals would be informative and may pave the way for identifying novel antifungal targets against this emerging pathogen.

# Methods

## Preparation of ¹³C, ¹⁵N-fungal cells

Five strains, including a *C. albicans* (SC5314) and five *C. auris* strains (AR386, I.3, P*ADH1*_KRE6a, P*ADH1*_KRE6b, kre6aΔ and kre6bΔ), were cultured in Yeast Nitrogen Base (YNB; DF0335159, Thermo Fisher Scientific) medium minus amino acids and ammonium sulfate, but supplemented with 6.0 g/L of ¹³C-glucose (CLM-1396-PK, Cambridge Isotope Laboratories) and 10.0 g/L of ¹⁵N-ammonium sulfate (NLM-713-PK, Cambridge Isotope Laboratories) to enable uniform labeling of fungal biomass. The medium was adjusted to pH 7.0, and cultures were incubated for 3 days at 30 °C. For *C. albicans* (SC5314) and *C. auris* (AR386), caspofungin (SML0425, Millipore Sigma) was added at 0.015 µg/mL and 0.05 µg/mL, respectively, below their minimum inhibitory concentrations (MIC), to enable the generation of labeled biomass. Other *Candida* strains were grown in the presence of caspofungin at their respective MIC levels. After growth, fungal materials were harvested by centrifugation at 7000 × g for 20 min. The collected pellets were thoroughly washed with 10 mM phosphate buffer (PBS, pH 7.4; P3999500, Thermo Fisher Scientific) to remove residual small molecules and reduce ion concentration. Approximately 100 mg of the whole-cell material was packed into a 4 mm magic-angle spinning (MAS) rotor for solid-state NMR characterization, while an additional 30 mg of material was packed into a 3.2 mm sapphire rotor for DNP experiments[91,100,101]. The native hydration level of the fungal cells was fully retained throughout the process.

## Transmission electron microscopy (TEM) imaging

Harvested *Candida* cultures were fixed in a solution of 2.5% (v/v) glutaraldehyde and 2% (w/v) paraformaldehyde (15700, Electron Microscopy Sciences) in 0.1 M PBS buffer (pH 7.4). After centrifugation, the suspensions were embedded in 3% agarose gel. The samples were rinsed with 0.1 M PBS (pH 7.4) and 0.05 M glycine, followed by postfixation with 2% osmium tetroxide (OsO₄; SKU 19152, Electron Microscopy Science) followed by three washes with deionized water. Dehydration was performed using a graded series of acetone (50%, 70%, 80%, 90%, and 100%) and propylene oxide in two 15 min cycles. Subsequent infiltration was carried out using a series of propylene oxide and Epon. Ultrathin sections were prepared for TEM analysis, with 1% uranyl acetate (22400, Electron Microscopy Sciences) En Bloc staining applied to enhance contrast. Images were acquired from perpendicular cross-sections of hyphae using a JEOL JEM-1400 electron microscope.

## Antifungal susceptibility testing

The antifungal drugs micafungin (SML2268, Millipore Sigma) and caspofungin were obtained as powder suspensions and dissolved in dimethyl sulfoxide (DMSO) for stock solutions of 10 mg/mL. Antifungal susceptibility testing was performed according to the European Committee on Antimicrobial Susceptibility Testing (EUCAST) protocol[102]. The growth was determined by measurement of absorbance at 600 nm using the LUMIstar Omega microplate reader (BMG LABTECH, Ortenberg, Germany) and expressed as percentage of the control growth (i.e. in the absence of the drug). In addition to microplate testing, a spot assay was used for further assessment. Ten-fold dilutions of yeast suspensions (ranging from $10^7$ to $10^4$ cells/mL) were spotted (5 µL per spot) onto solid yeast extract peptone dextrose (YEPD) plates with or without the antifungal drugs. Plates were incubated at 37 °C for 24 h to evaluate drug effectiveness.

## *KRE6a/b* genetic constructs

The kre6abΔ, kre6aΔ and kre6bΔ constructs were generated by fusion PCR of ~500-bp upstream and downstream sequences of the targeted genes flanking the *NAT1* cassette (nourseothricin resistant) derived from plasmid pJK795[103], as illustrated in Supplementary Figs. 7–9. For complementation strains, the *KRE6a* and *KRE6b* genes were reintroduced at their native locus using specific RNAs guides in order to replace the previously inserted *NAT1* cassette. For this purpose, we generated by fusion PCR a construct where the promoter, the ORF of the gene (*KRE6a* or *KRE6b*), and its terminator region were fused to the *BleMX* cassette (from plasmid pCdOpt-BMX)[104] used as selection marker for bleomycin resistance, as illustrated in Supplementary Figs. 10–12. Because the *KRE6a* and *KRE6b* loci are separated by only 764 bp, the *BleMX* cassette was inserted at -426 bp from the start codon of *KRE6b* in order to preserve a sufficient part of the *KRE6a* terminator (338 bp) and *KRE6b* promoter (426 bp).

For overexpression of *KRE6a* and *KRE6b*, we generated two strains, P*ADH1*_KRE6a and P*ADH1*_KRE6b, respectively, where the gene was placed under the control of the constitutive *ADH1* promoter and integrated in the *C. auris* neutral site (*CauNi*). The PCR product of the gene (*KRE6a* or *KRE6b*) was cloned at KasI/NruI sites of the plasmid pjli8 containing the *ADH1* promoter, the *ACT1* terminator, the *SAT1* marker (nourseothricin resistant) and the *CauNi* locus[105], as illustrated in Supplementary Fig. 13. The plasmid was linearized by StuI for transformation. Transformations were performed by a CRISPR-Cas9 approach via electroporation[106]. Gene-specific guide RNA was designed to contain 20-bp homologous sequences of the target region for integration of the genetic constructs. The mix of DNA fusion PCR products, RNA guides, Cas9 endonuclease 3NLS (Integrated DNA Technologies Inc., Coralville, IA), and tracrRNA (a universal transactivating CRISPR RNA) were prepared[107]. Transformants were selected at 37 °C on YEPD containing 200 µg/mL of nourseothricin (Werner BioAgents, Jena, Germany). Correct integration of the constructs was verified by PCR. All primers and guide RNA are listed in Supplementary Table 8.

## Real-time reverse transcription (RT)-PCR

*C. auris* strains were grown overnight in liquid YEPD (37 °C, constant agitation). The resulting culture broth was adjusted to a concentration of $0.75 \times 10^7$ cells/ml in 5 ml of YEPD and further incubated for 2 h to reach a final concentration of $1.5 \times 10^7$ cells/ml. RNA was extracted with the Quick-RNA fungal/bacterial miniprep kit (Zymo Research, Freiburg im Breisgau Germany) and purified with DNase using the DNA-free kit (Thermo Fisher Scientific Inc., Waltham, MA). RNA concentration was measured with NanoDrop 1000 instrument (Witec AG, Sursee, Switzerland) and samples were stored at −80 °C. For each strain/condition, 1 µg of RNA was converted into cDNA using the Transcriptor high-fidelity cDNA synthesis kit (Roche, Basel, Switzerland). The PCR mix contained 10 ng cDNA, 10 µl PowerUp SYBR green master mix (Applied Biosystems, Waltham, MA), 300 nM of each primer and nuclease-free water for a total volume of 20 µl for each reaction. Primers used for amplification of *ACT1*, *KRE6a*, *KRE6b*, *FKS1* and *CHS1* are listed in Supplementary Table 8. Real-time PCR was performed with the QuantStudio (Thermo Fisher Scientific Inc., Waltham, MA) including a melt curve stage using the following program: 10 min activation at 95 °C, 15 sec denaturation at 95 °C and 1 min annealing/extension at 60 °C (40 cycles). The $2^{-\Delta\Delta CT}$ method was used for calculation of the target gene expression normalized to *ACT1* expression[107,108]. Results were expressed as fold change of the gene expression of the reference strain (AR386). Statistical analyses were performed using the *t*-test method (GraphPad).

## ¹³C solid-state NMR analysis of cell wall biomolecules

Most solid-state NMR experiments were performed on an 800 MHz (18.8 Tesla) Bruker Avance Neo spectrometer equipped with a 3.2-mm MAS HCN probe, except for the relaxation and water-edited experiments, which were conducted on a Bruker Avance 400 MHz (9.4 Tesla) spectrometer using a 4 mm MAS HCN probe. All experiments were acquired under MAS frequencies of 10–13.5 kHz and a temperature of 290 K. ¹³C chemical shifts were externally referenced to the adamantane CH₂ peak at 38.48 ppm on the tetramethylsilane (TMS) scale.

Unless otherwise specified, typical radiofrequency field strengths were 80-100 kHz for $^1$H decoupling, 62.5 kHz for $^1$H hard pulses and 50-62.5 kHz for $^{13}$C. $^1$H-$^{13}$C CP was achieved using a 1-ms 50 kHz spinlock square pulse. Key experimental parameters were provided in Supplementary Table 9.

1D $^{13}$C spectra were acquired using various polarization methods to selectively detect the rigid and mobile components of the fungal molecules. Rigid components were identified through 1D $^{13}$C CP with a 1 ms contact time. Quantitative detection of all molecules was achieved using 1D $^{13}$C direct polarization (DP) with a long (30 s) recycle delay, while shortening this recycle delay to 2 s allowed us to selectively detect mobile molecules. Additionally, a 1D refocused J-INEPT experiment was performed to measure highly dynamic components[109]. 1D $^{13}$C CP spectra were repeatedly acquired before and after 2D and 3D experiments to closely monitor sample integrity, and the $^{13}$C spectra are highly reproducible (Supplementary Fig. 16). The fungal material remaining in the MAS rotor after the experiment can be used to initiate new cultures, indicating that a substantial portion of the cells remain viable post-experiment (Supplementary Fig. 17). A new batch was prepared if any changes were detected. Samples were stored at -20 °C freezer when not in use.

Two-dimensional (2D) correlation experiments were performed using standard techniques recently applied to fungal cell walls[52,56,57,110]. These techniques included through-space correlation methods, such as proton-assisted recoupling (PAR)[111,112], combined R2$_n^v$-driven spin diffusion (CORD)[113], and radiofrequency-driven recoupling (RFDR)[114,115] pulse sequences. For *Candida* cells, the CORD experiment was performed at ambient temperature for resonance assignment. CORD efficiently drives homonuclear $^{13}$C-$^{13}$C polarization transfer under fast MAS conditions and high magnetic fields[113]. At the slower MAS rate of 13.5 kHz used in this study, its performance is comparable to that of Dipolar-Assisted Rotational Resonance (DARR), while offering the advantage of lower radiofrequency power requirements during the recoupling period. Compared to proton-driven spin diffusion (PDSD), CORD provides higher efficiency due to the use of active recoupling pulses[113]. Additionally, through-bond polarization transfer techniques such as refocused J-INADEQUATE and refocused J-INEPT were employed[109,116]. 2D refocused DP J-INADEQUATE spectra were acquired using four tau periods of 2.3 ms each, accompanied by short recycle delays of 2 s to detect mobile components exhibiting through-bond connectivity. The 2D refocused $^1$H-$^{13}$C INEPT experiment was conducted with two delay intervals of 1/(4J$_{CH}$) each, followed by two additional delay intervals of 1/(6J$_{CH}$) each, where J$_{CH}$ represents the J-coupling constant, set to 145 Hz. A total of 256 transients were recorded with recycle delays of 4 s. The combined application of these methods enables the assignment of NMR resonances for various carbohydrate components, with chemical shifts tabulated in Supplementary Table 1. facilitates differentiation of their presence across different dynamic regimes, and enhances the analysis of their polymorphic structure, composition, and the physical packing of biomolecules within native cell walls.

Measurement of $^{13}$C-T$_1$ relaxation was achieved using the Torchia-CP experimental scheme[117], with z-filter durations ranging from 0.1 µs to 12 s. For each resolved peak, the intensity decay was monitored as the z-filter time increased, and the data were fit to a single exponential function to determine the $^{13}$C-T$_1$ relaxation time constants. The absolute intensity was pre-normalized based on the number of scans for each spectrum. For measuring $^{13}$C-detected $^1$H-T$_{1\rho}$ relaxation, a Lee-Goldburg (LG) spinlock sequence combined with LG-CP was employed[118]. This technique effectively suppressed $^1$H spin diffusion during both the spinlock and CP contact periods, allowing for site-specific determination of the $^1$H-T$_{1\rho}$ relaxation times of proton sites through the

detection of their directly bonded $^{13}$C carbons. The decay in peak intensity was modeled using a single exponential function to extract the $^1$H-T$_{1\rho}$ time constants. All relaxation curves were analyzed and fit using OriginPro 9.

To assess the water accessibility of polysaccharides, water-edited 2D $^{13}$C-$^{13}$C correlation spectra were acquired[119–122]. The experiment commenced with $^1$H excitation, followed by a $^1$H-T$_2$ filter of 3.0 ms × 2 for the apo *C. auris* sample and 5.0 ms × 2 for the caspofungin-treated *C. auris* sample, which eliminated 97% of the polysaccharide signals while retaining 80% of the water magnetization. Subsequently, the water magnetization was transferred to the polysaccharides using a 4 ms $^1$H mixing period, followed by a 1 ms $^1$H-$^{13}$C cross-polarization (CP) for site-specific detection of $^{13}$C signals. A 50 ms DARR mixing period was employed for both the water-edited spectrum and a control 2D spectrum, which displayed the full intensity. The relative intensity ratio between the water-edited spectrum and the control spectrum was quantified for all cell wall samples to reflect the extent of hydration. The intensities were pre-normalized based on the number of scans for each spectrum.

## $^1$H-detected solid-state NMR

The rigid molecules in apo *C. albicans* and *C. auris* cell walls were detected using a CP-based 2D hCH experiment[123] on a 600 MHz Bruker Avance Neo spectrometer using a 1.3 mm triple-resonance HCN probe at 60 kHz MAS. A small amount of sodium trimethylsilypropane-sulfonate (DSS) dissolved in D$_2$O was added to the sample for temperature calibration and $^1$H chemical shift referencing, with the DSS peak set to 0 ppm. To reduce the frictional heating, the thermocouple reported temperature was set to 255 K, and the calibrated sample temperature was 304 K for *C. albicans* and 300 K for *C. auris*. A 90° pulse for $^1$H was applied at 100 kHz rf (2.5 µs), and the CP conditions were optimized with radiofrequency (rf) power of 46.8 kHz for $^1$H and 10.9 kHz for $^{13}$C with a 90–100% ramp on the proton channel. During t$_1$ evolution, slpTPPM decoupling[124] at 20.28 kHz rf was applied to the $^1$H channel, and WALTZ-16 decoupling was used on both the $^{13}$C and $^{15}$N channels with rf power of 10 kHz. Water suppression was achieved with the MISSISSIPPI sequence[125], with 30 kHz rf power for 100 ms. The total experimental time for each sample was 11.3 h. The experimental parameters were documented in Supplementary Table 10.

Resonance assignment of mobile carbohydrates were achieved using proton-detected 3D (as well as its 2D version) $^1$H-$^{13}$C J-hCCH TOCSY[126,127] with DIPSI-3 mixing[128]. This experiment was utilized by Bahri et al. for assigning side-chain resonances of proteins and analyzing the mobile molecules of the fungal cell walls[73]. The experiments were performed on an 800 MHz Bruker Avance Neo NMR spectrometer with a 3.2 mm triple-resonance HCN MAS probe at 13.5 kHz MAS. The sample temperatures were 296 K for *C. albicans* was 295 K for *C. auris*. The 90° pulse widths for $^1$H and $^{13}$C were 3.5 µs (71.429 kHz) and 5 µs (50 kHz), respectively. SPINAL-64 decoupling[129] was applied during both t$_1$ and t$_2$ evolutions with a rf filed strength of 71.4 kHz, WALTZ-16 decoupling was applied on $^{13}$C channel with 17 kHz rf power during $^1$H detection[130]. Water suppression was achieved with the MISSISSIPPI sequence[125] using 26 kHz rf for 40 ms. The broadband DIPSI-3 mixing was applied to obtain $^{13}$C-$^{13}$C homonuclear correlations, with a 2 ms spin-lock pulse. The mixing period was set to 25 ms and a rf field strength of 17 kHz was applied for both DIPSI-3 and for the spin-lock pulse (SLx). 128 TD points were recorded for both t$_1$ and t$_2$ evolution, with 8 transients co-added per point and a 1.89 s recycle delay. The total experimental time was 3.1 d for the 3D experiment. A 2D hcCH TOCSY with DIPSI-3 mixing was also performed using the same parameters as the 3D experiment but with t1 set to 1 TD point and t2 to 512 points. The recycle delay was set to 2 s. This 2D experiment took 2.5 h.

## MAS-DNP sample preparation and experiments

MAS-DNP experiments were conducted using a freshly prepared stock solution of 10 mM AsymPolPOK (C015P01, Cortecnet) in a $d_8$-glycerol/$D_2O$/$H_2O$ (60/30/10 Vol%) mixture[83,91,100,131]. Approximately 30 mg of $^{13}C$, $^{15}N$-labeled *C. auris* cells were combined with 50 μL of the stock solution and manually ground for 10 min to facilitate radical penetration into the porous cell walls. The sample was transferred to a 3.2 mm sapphire rotor for DNP measurements. All experiments were carried out on a 600 MHz/395 GHz MAS-DNP spectrometer at the National High Magnetic Field Laboratory (Tallahassee, FL) with an 89 mm bore and gyrotron microwave source[132]. Spectra were acquired using a 3.2 mm HCN probe at 8 kHz MAS and 100 K. Gyrotron cathode currents ranged from 130–150 mA, and the voltage was set to 16.2 kV, which corresponded to 10–13 W at the probe base. The NMR sensitivity enhancement ($\varepsilon_{on/off}$) was ~57 and the DNP signal buildup time ranged between 5.2 and 13 s. 2D $^{13}C$-$^{13}C$ spectra were collected with PAR recoupling durations of 5 ms and 20 ms for both apo and caspofungin-treated *C. auris* AR386 strain samples. The PAR duration included $^{13}C$ and $^1H$ irradiation frequencies set to 56 kHz and 53 kHz, respectively.

## Reporting summary

Further information on research design is available in the Nature Portfolio Reporting Summary linked to this article.

## Data availability

All the original ssNMR data files generated in this study have been deposited in the Zenodo repository under the number [https://doi.org/10.5281/zenodo.15594130]. Unless otherwise stated, all data supporting the results of this study can be found in the article, supplementary, and source data files. Source data are provided as a Source Data file. Source data are provided with this paper.

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

## Acknowledgements

This work was primarily supported by the National Institutes of Health (NIH) grant R01AI173270. F.L. and J.L. are supported by the Swiss National Science Foundation (SNSF, project number 310030_192611) and by the Santos-Suarez foundation. A portion of this work was performed at the National High Magnetic Field Laboratory, which is supported by National Science Foundation Cooperative Agreement No. DMR-2128556 and the State of Florida. The MAS-DNP system at NHMFL is funded in part by NIH RM1-GM148766. F.J.S. was funded by the post-doctoral scholar award from the Provost's Office at Florida State University. N.A.R.G. acknowledges support of Wellcome Trust Investigator, Collaborative, Equipment, Strategic and Biomedical Resource awards (101873, 200208, 215599, 224323). N.A.R.G. also thanks the MRC (MR/M026663/2) and the MRC Centre for Medical Mycology (MR/N006364/2) for support. This study/research is funded by the National Institute for Health and Care Research (NIHR) Exeter Biomedical Research Centre (BRC). The views expressed are those of the author(s) and not necessarily those of the NIHR or the Department of Health and Social Care. The authors thank Dr. Jean-Paul Latge for helpful discussions.

## Author contributions

M.C.D.W., K.S., and Y.X. conducted 13C solid-state NMR experiments. J.R.Y. conducted 1H solid-state NMR experiments. M.C.D.W., K.S., Y.X. and P.W. prepared fungal samples. F.J.S. and F.M.-V. conducted MAS-DNP experiments. M.C.D.W., K.S., Y.X., J.R.Y. and T.W. analyzed the experimental data. K.S. conducted the TEM and SEM measurements. J.L. and F.L. conducted gene editing of *C. auris* and echinocandin susceptibility test. All authors wrote the manuscript. P.W., N.A.R.G., F.L., and T.W. designed and supervised the project.

## Competing interests

The authors declare no competing interests.
