## [Transparent Peer Review file · Nature Communications]

Distinct Echinocandin Responses of *Candida albicans* and *Candida auris* Cell Walls Revealed by Solid-State NMR

Corresponding Author: Professor Tuo Wang

Version 0:

Reviewer comments:

Reviewer #1

(Remarks to the Author)

Effectiveness in treating candidiasis can be impaired by drug resistance in both *Candida albicans* and the recently emerged pathogen *Candida auris*, the latter exhibiting natural resistance. The authors note that besides resistance, *Candida* cells can become tolerant to echinocandins, one of the groups of anti-*Candida* drugs. Resistance in *C. albicans* can arise from mutations in the B-1,3-glucan synthase FKs subvariant. Tolerance seems to arise through the emergence of sub populations surviving drug- induced distress. Tolerance is associated with increases in exposed chitin accompanied by decreases in B-glucans in the cell wall. The cell wall of *Candida albicans* is composed of an inner layer of chitin, B-1,3-glucan and B-1,6-glucan, and an outer layer of mannan proteins, a significant proportion of which are linked to the glucan-chitin inner layer. The authors argue that the high mortality rates of *C. auris* infections, and tolerance in particular, and the frequent ineffectiveness of the present antifungals, necessitates the development of a new drugs, which will be facilitated by elucidation of the mechanisms of resistance and drug-induced tolerance. The latter in particular may provide new targets for drug development. To this end they have applied NMR in an analysis of *C. auris* and *C. albicans*, techniques applied to a number of other fungal cell walls, most notably those of *Aspergillus* species, in previous studies of morphotypes and stress responses. They demonstrate that while *C. auris* and *C. albicans* share general wall composition, the two species differ markedly in the changes in cell wall architecture in response to low levels of caspofungin associated with tolerance. Elucidating the differences in the mechanisms at play will assist in identifying new targets for drug development. This reviewer agrees with this assertion and believes that this study represents an excellent step in this pursuit. Although not an expert in NMR and DNP technologies, this reviewer is familiar with the biology and pathology of *Candida* species, and to a lesser degree with drug resistance, and from that context and point of view, has critiqued the paper with the goal of making it more accessible to an audience interested in drug resistance and tolerance of *C. auris*, but with little or no expertise in NMR and DNP. Suggestions and queries follow.

Pages 5-7. First, the authors don't explain why they detail the wall architecture and composition of *A. fumigatus* in the Results section for comparison with *C. albicans*. I think the takeaway message is that the two fungal genera form quite different walls and the *C. albicans* cell wall is simpler than the *A. fumigatus* wall. It would seem reasonable to have the *A. fumigatus* results in a separate communication that could be referred to. Alternatively, the authors could explain why they present the detailed *Aspergillus* results with a few sentences at the beginning of the first section of RESULTS. It seems to this reviewer that the comparison of the wall of *A. fumigatus* with the wall of *C. albicans* was added to support the results in the next section that the overall wall composition of *C. auris* and *C. albicans* are similar, but dissimilar to that of *Aspergillus* and *Candida*. This reader, after reading the first section of results, might expect the *Aspergillus* data to be tied into the mechanism of resistance and tolerance of the *Candida* species to echinocandins.

Lines 118-122. It is confusing to this reviewer that the authors have compared the cell walls of the filamentous form of *A. fumigatus* and the yeast form of *Candida*, since *Candida* species form yeast and filamentous morphologies. The risk is that the differences between *Candida* and *Aspergillus* may not be as great if the filamentous form of *Candida* was compared to the filamentous form of *Aspergillus*. If the authors have performed a comparison of the filamentous form of *Candida*, it would be quite useful in excluding this possibility.

Figure 1 a. The SEMs of *C. albicans* cells are of poor quality and exhibit distortions in the wall suggesting sub optimum preparation conditions. Also, the size of some of the cells appear quite large. The SEMs of *A. fumigatus* show only branching filaments, but the diagram shows hyphae and conidiophores. Did the *A. fumigatus* preps have multiple cell types? That could be problematic.

Figure 1 a. It would also be helpful to show SEMs of the preparations of caspofungin- treated *C. albicans* and *C. auris*.

Lines 172-178. It would be helpful if the authors first explained the difference between “rigid” and “flexible” side chains and the respective roles of beta-1, 3-glucan and beta-1, 6- glucan side chains. In a similar fashion, the difference between mannan fibrils and chitin fibrils should be introduced in regard to “rigid” versus “dynamic,” perhaps at the beginning of the section (line 17), so the reader knows why the results that follow are important.

Lines 117 to 178. It also seems relevant to explain why the results on rigidity may be related to drug resistance and tolerance in the Introduction or early in the Results section before you present your detailed results on glucan branching.

Lines 232-235. The authors should test for significance of the caspofungin effects on cell wall thickness for *C. albicans* and *C. auris*, providing p values. For the two strains of *C. auris* the differences are probably not significant, so the statement that there were “minor changes” may not be accurate.

Line 236. The use of the phrase “microscopic-level changes” should be deleted.

Line 362. Should change “microscopic differences” to differences in “wall thickness.”

Figure 3. It would be very helpful for *Candida* scientists unfamiliar with wall architecture if the walls were diagrammed for untreated and treated cells, noting in the diagrams the changes observed for both *Candida* species. This would supplement Figure 3e, which shows changes in content, but not architecture. This might be helpful for the discussion beginning in line 279.

Lines 309-310. I’m not sure if it is necessary to provide a “vide infra” statement here.

Lines 381-383. This sentence summarizes the results of the preceding experiments on chitin-associated beta-glucans, but, as with other analyses in the Results section, does not explain its importance in understanding how caspofungin affects tolerance and resistance. This reviewer suggests that this analysis, as well as preceding analyses, should begin in each section with an explanation of why the following analysis is important within that context.

The Discussion section is excellent. It reviews the extensive number of important observations that lay the groundwork for eventually pinpointing new targets for antifungal drug discoveries. Subsequent experimental approaches based on these observations are explicitly proposed and are no doubt in progress.

The suggestions I have made may help make the paper more amenable to an audience versed in the biology of *Candida* species and drug development, but not well versed in the advanced technologies employed in the study and not well versed in the fungal wall literature. Most notably, I suggest considering diagrams of the cell wall that describe the inner and outer layers of the wall, that distinguish or define rigid scaffolds from mobile or flexible components, that distinguish hydrated from dehydrated components, that list the components and their characteristics in the inner and outer layers, and that describe the differences in the responses in the wall to caspofungin between *C. albicans*, *C. auris*, AR-386 and *C. auris*. I. 3, in the context of susceptibility, tolerance and resistance. In summary, this is an excellent, in-depth study that will impact the field.

Reviewer #2

(Remarks to the Author)

In this study, Widanage and colleagues investigated if the cell wall structures of *Candida auris* and *Candida albicans* were impacted by echinocandin treatment. The authors used complementary approaches, including solid-state NMR analyses. Overall, the authors found that the cell walls of both *Candida auris* and *Candida albicans* are similar. However, only *Candida albicans* responded by inducing cell wall thickening and changes in chitin and beta-1-3-glucan dynamics. This study provides a molecular-level understanding of the mechanisms that these pathogens use to overcome echinocandin treatment, which may be useful in future drug development studies.

My comments and suggestions are below:

SC5314 and AR286 have similar cell wall changes after caspofungin treatment, but I.3 is the most different. Can you expand on what contributes to the difference of I.3 and AR286 cell walls although they are both *C. auris*?

Consider providing evidence or past studies that support that the biradical used for DNP does not change the cell wall architecture.

How long are the NMR experiments (this information was only provided for some experiments)? This is an important consideration in case the cells are changing over the course of very long experiments. Consider providing evidence that the cell wall compositions and architectures are the same at the beginning and end of the NMR experiments.

Are the cells viable and growing normally after the solid-state NMR experiments? This would lend support to that the cell wall architecture, including levels of hydration, are not impacted during the solid-state NMR experiments and that the spectra that you report reflect native conditions.

beta-1,6-glucans and beta-1,3-glucans can compensate for one another. Which Kre6 proteins compensate for the similar or higher levels of beta-1,6-glucans? Can you provide more clarifications with citations?

Consider including a brief statement describing your choice of CORD (example: Figure 1) instead of another similar pulse sequence such as PDS or DARR.

There seems to be a typo in the last sentence of the Figure 2 legend about the experimental parameters for the hcCH and hCH spectra.

There seems to be a typo in line 315, since I believe that you are referencing Figure 4A (not Fig 3A).

Reviewer #3

(Remarks to the Author)

This manuscript by Widanage, Singh, and Li et al. focuses on a detailed analysis of the cell wall structure of two fungal pathogens, *Candida albicans* and *Candida auris*. The analysis is similar to that performed for other fungal pathogens in response to stress, including *Aspergillus*, *Cryptococcus*, *Schizophyllum*, *Neurospora*, and *Mucor*. The major finding is that the *Candida* species characterized share a similar cell wall composition that is distinct from other fungi and show changes in response to echinocandin exposure. Although the cell wall analysis may present an incremental advance, there are a number of issues with the manuscript. First, the stated goal of the manuscript of asking how echinocandins remodel cell wall structures, thereby reducing drug effectiveness is very strange. The echinocandins inhibit a key cell wall linker molecule as part of their mode of action and the impact on the cell wall is the basis of their effectiveness. The premise upon which it is inferred that cell wall changes induced by the echinocandins are the basis of reduced echinocandin efficacy is entirely unclear. Second, the conclusion that profound alterations induced by caspofungin in *Candida* cell wall architecture suggest that cell wall structural contribute substantially to drug resistance and tolerance is not supported by the data presented. An observation of cell wall changes accompanying drug treatment does not implicate cell wall changes in resistance. Additional issues are addressed in the specific points below.

Specific Points:

1. The background that echinocandin tolerance arises through the growth of subpopulations of cells that survive despite drug-induced stress cites two references, one of which focuses on azoles not echinocandins and the other is a broad review article. This point needs to be appropriately supported by the literature.
2. If the goal of the study is to understand tolerance, then there should be analysis of isogenic cells that acquired tolerance relative to their sensitive parental strains.
3. For the comparative analysis shown in Figure 1, were the *Aspergillus* cell walls prepared in the same way as the *Candida* cell walls? If not, could the methodology affect the results? Were the *Aspergillus* results previously published (in which case this would need to be addressed).
4. Only one *Candida albicans* strain is used as representative of the entire species, despite large phenotypic variation well documented across isolates.
5. There is no rationale provided for the specific concentration of echinocandin used for each species. Different effects could be attributable to different effective concentrations of echinocandins with no analysis at different doses. If the dose of caspofungin has minimal antifungal effect, as with *Candida auris*, it would be consistent that there is minimal effects on the cell wall.
6. To interpret the phenotype of a gene deletion mutant and rule out the possibility of phenotypic effects being attributable to spurious mutations, it is critical to generate a complemented strain.
7. The authors conclude FKS1 is overexpressed in the *kre6a* mutant but the data shown is questionable – the magnitude of effect is very small and unlikely to contribute any increase in resistance.

Minor Points

Lines 504-506: This statement needs to be refined as not all *Candida* species transition from yeast to filamentous growth
Line 26: implicating not the correct word
Line 63: Echinocandin should not be capitalized mid sentence
Line 65-66: "However, echinocandin treatment often results only in partial inhibition of β -1,3-glucan synthesis" Requires a reference.

Version 1:

Reviewer comments:

Reviewer #1

(Remarks to the Author)

Review of revisions by Widanage et al.

The authors have revised the manuscript in response to all of the suggestions I articulated in the original review. These changes are outlined below.

1. As I suggested, the *A. fumigatus* results were interesting but served no purpose in comparative analysis of *Candida* species and the basis of the drug susceptibility differences between the latter. As recommended, the authors removed these results. This included removal of Figures 1a, c, d and f, Table 1, supplemental Figure 1, and the first section of Results.

2. I suggested clarifying the molecular differences between rigid and mobile layers of the cell wall. This has been dealt with adequately in the first section of Results.

3. The authors have added, as suggested, p-values for the differences between apo and drug tested drug-treated samples, demonstrating significance.

4. As suggested, the authors have adequately explained the relevance of their observation of the consistent structure of beta-1, 3-glucan bound by chitin in the apo and caspofungin-treated cell wall of *C. auris* in original lines 381-383.

5. Finally, the authors have added, as suggested, a new Figure 8 which diagrams for both *Candida* species the cell wall without and with caspofungin.

I might also point out that the authors analyzed complemented strains of the KRE mutants in response to the suggestion by reviewer 3, a point I had considered, but forgot to add to my original review.

I commend the authors for attending to the revisions and suggestions rather than rebutting the suggestions made by the three reviewers.

The editorial board may name me if the paper is published.

David R. Soll

Carver/Witschi Professor, Emeritus
University of Iowa

Reviewer #2

(Remarks to the Author)

Thank you for addressing my prior comments. I have no additional comments, suggestions, or concerns.

Reviewer #3

(Remarks to the Author)

The authors have sufficiently addressed all my original critiques and I now recommend this manuscript be accepted to Nature Communications.

Responses to Reviewers' Comments

Reviewer #1:

Effectiveness in treating candidiasis can be impaired by drug resistance in both *Candida albicans* and the recently emerged pathogen *Candida auris*, the latter exhibiting natural resistance. The authors note that besides resistance, *Candida* cells can become tolerant to echinocandins, one of the groups of anti-*Candida* drugs. Resistance in *C. albicans* can arise from mutations in the B-1,3-glucan synthase FKs subvariant. Tolerance seems to arise through the emergence of sub populations surviving drug-induced distress. Tolerance is associated with increases in exposed chitin accompanied by decreases in B-glucans in the cell wall. The cell wall of *Candida albicans* is composed of an inner layer of chitin, B-1,3-glucan and B-1,6-glucan, and an outer layer of mannan proteins, a significant proportion of which are linked to the glucan-chitin inner layer. The authors argue that the high mortality rates of *C. auris* infections, and tolerance in particular, and the frequent ineffectiveness of the present antifungals, necessitates the development of a new drugs, which will be facilitated by elucidation of the mechanisms of resistance and drug-induced tolerance. The latter in particular may provide new targets for drug development. To this end they have applied NMR in an analysis of *C. auris* and *C. albicans*, techniques applied to a number of other fungal cell walls, most notably those of *Aspergillus* species, in previous studies of morphotypes and stress responses. They demonstrate that while *C. auris* and *C. albicans* share general wall composition, the two species differ markedly in the changes in cell wall architecture in response to low levels of caspofungin associated with tolerance. Elucidating the differences in the mechanisms at play will assist in identifying new targets for drug development. This reviewer agrees with this assertion and believes that this study represents an excellent step in this pursuit. Although not an expert in NMR and DNP technologies, this reviewer is familiar with the biology and pathology of *Candida* species, and to a lesser degree with drug resistance, and from that context and point of view, has critiqued the paper with the goal of making it more accessible to an audience interested in drug resistance and tolerance of *C. auris*, but with little or no expertise in NMR and DNP. Suggestions and queries follow.

We appreciate the positive comments from the reviewer regarding the significance of this study and the constructive critiques, which have been addressed as detailed below to improve the manuscript.

Pages 5-7. First, the authors don't explain why they detail the wall architecture and composition of *A. fumigatus* in the Results section for comparison with *C. albicans*. I think the takeaway message is that the two fungal genera form quite different walls and the *C. albicans* cell wall is simpler than the *A. fumigatus* wall. It would seem reasonable to have the *A. fumigatus* results in a separate communication that could be referred to. Alternatively, the authors could explain why they present the detailed *Aspergillus* results with a few sentences at the beginning of the first section of RESULTS. It seems to this reviewer that the comparison of the wall of *A. fumigatus* with the wall of *C. albicans* was added to support the results in the next section that the overall wall composition of *C. auris* and *C. albicans* are similar, but dissimilar to that of *Aspergillus* and *Candida*. This reader, after reading the first section of results, might expect the *Aspergillus* data to be tied into the mechanism of resistance and tolerance of the *Candida* species to echinocandins.

We appreciate the insightful comment, which aligns with feedback from other reviewers, all highlighting confusion arising from the comparison between *Candida* species and *Aspergillus*. To address this concern, we have removed all elements related to this comparison. Specifically, we have removed the original Fig. 1a, c, d, f, as well as the original Supplementary Figure 1. Furthermore, we have completely rewritten the first subsection of the Results section to provide a focused description of *C. albicans* cell wall components, and their distribution in dynamically distinct structural domains, rather than including comparisons across different species.

Lines 118-122. It is confusing to this reviewer that the authors have compared the cell walls of the filamentous form of *A. fumigatus* and the yeast form of *Candida*, since *Candida* species form yeast and filamentous morphologies. The risk is that the differences between *Candida* and *Aspergillus* may not be as great if the filamentous form of *Candida* was compared to the filamentous form of *Aspergillus*. If the authors have performed a comparison of the filamentous form of *Candida*, it would be quite useful in excluding this possibility.

Thanks and we fully agree that it is not a fair comparison. As detailed in the responses to the last question. We have now removed the comparison to *A. fumigatus* but instead streamlined the first section of the Results to fully focus on *Candida* cell wall polysaccharides.

Figure 1 a. The SEMs of *C. albicans* cells are of poor quality and exhibit distortions in the wall suggesting sub optimum preparation conditions. Also, the size of some of the cells appear quite large. The SEMs of *A. fumigatus* show only branching filaments, but the diagram shows hyphae and conidiophores. Did the *A. fumigatus* preps have multiple cell types? That could be problematic. Figure 1 a. It would also be helpful to show SEMs of the preparations of caspofungin- treated *C. albicans* and *C. auris*.

The original SEM figures were intended to help general readers without a background in fungal biology to understand the morphological differences between *A. fumigatus* mycelia and *Candida* yeast cells. However, as we acknowledged in our earlier responses, this comparison is neither appropriate nor necessary. We fully agree with the reviewer that these images are more distracting than informative in the context of this study. Accordingly, we have removed all SEM images of *C. albicans* and *A. fumigatus*, as well as the original Supplementary Figure 1 and Table 1. This revision allows us to focus more directly on the NMR findings, which are central to the study.

Lines 172-178. It would be helpful if the authors first explained the difference between “rigid” and “flexible” side chains and the respective roles of beta-1, 3-glucan and beta-1, 6- glucan side chains. In a similar fashion, the difference between mannan fibrils and chitin fibrils should be introduced in regard to “rigid” versus “dynamic,” perhaps at the beginning of the section (line 117), so the reader knows why the results that follow are important.

Thanks for the helpful advice. Enhanced rigidity is a typical indicator of large or ordered aggregates, such as chitin microfibrils, whereas molecules that are spatially separated from these mechanical cores are typically mobile and hydrated. We have now explained the rigid/dynamic characteristics in the first subsection of the Results with the following statements: “These components exhibit reduced molecular mobility due to the formation of ordered or semi-crystalline domains, or tightly packed aggregates that enhance the stiffness of the cell wall,” and “Positioned within the soft mesh and spatially distinct from the rigid core, these mobile molecules generally remain highly flexible and well-hydrated.”

Meanwhile, we have also refined the description of the structural relevance of these observations in the last paragraph of the first subsection of the Results. It now reads as follows: “These findings demonstrate that *Candida* cell walls exhibit a relatively simple structure, with the rigid structural framework containing only chitin microfibrils and stiff linear β -1,3-glucan backbones, which may primarily adopt the triple helix conformation, within the inner layer. These rigid components are supported by more flexible β -1,6-glucans and a small portion of branched β -1,3-glucans of the glucan complex. Both β -1,6-glucans and branched β -1,3-glucans have a branching topology and may exhibit a reduced degree of intermolecular association compared to chitin microfibrils and β -1,3-glucan triple helices. Mannan forms fibrillar structures that particularly dominate the outer layer of the cell wall, and our findings indicate that these fibrils are dynamically fluctuating, in contrast to the chitin microfibrils that exhibit partial structural order and crystallinity, as well as stiffness.”

Lines 117 to 178. It also seems relevant to explain why the results on rigidity may be related to drug resistance and tolerance in the Introduction or early in the Results section before you present your detailed results on glucan branching.

We have now added a brief explanation to the end of the first paragraph of the Results section that “These mobile domains regulate water accessibility and influence the diffusion of soluble molecules, including antifungals, while rigid, often partially dehydrated structural domains serve as a physical barrier, reducing overall permeability. Recent observations suggest that the balance between rigid structural components and mobile, water-accessible regions allows fungi to fine-tune cell wall permeability, aiding adaptive survival.”

Lines 232-235. The authors should test for significance of the caspofungin effects on cell wall thickness for *C. albicans* and *C. auris*, providing p values. For the two strains of *C. auris* the differences are probably not significant, so the statement that there were “minor changes” may not be accurate.

Thanks. We have now included the p-values in Figure 3b and updated the corresponding figure caption accordingly: “Statistically significant: *p-value \leq 0.05. The p-values are expressed for the comparisons between apo and drug-treated samples for each strain.” The original statement was correct following the t-test results.

Line 236. The use of the phrase “microscopic-level changes” should be deleted.

We deleted “microscopic-level” from this sentence.

Line 262. Should change “microscopic differences” to differences in “wall thickness.”

We have now changed it to “...despite their differences in wall thickness”

Figure 3. It would be very helpful for *Candida* scientists unfamiliar with wall architecture if the walls were diagrammed for untreated and treated cells, noting in the diagrams the changes observed for both *Candida* species. This would supplement Figure 3e, which shows changes in content, but not architecture. This might be helpful for the discussion beginning in line 279.

We have now added a new **Figure 7** to show the schematic illustrations of the *C. albicans* and *C. auris* AR386 cell walls. We also referred to this figure in the Discussion section to help convey the structural concepts.

Lines 309-310. I’m not sure if it is necessary to provide a “vide infra” statement here.

Thanks. “Vide infra” has been deleted.

Lines 381-383. This sentence summarizes the results of the preceding experiments on chitin-associated beta-glucans, but, as with other analyses in the Results section, does not explain its importance in understanding how caspofungin affects tolerance and resistance. This reviewer suggests that this analysis, as well as preceding analyses, should begin in each section with an explanation of why the following analysis is important within that context.

We appreciate the advice. We have now modified this subsection to start with “To investigate whether caspofungin treatment affects the spatial organization of polysaccharides within the mechanical core of *C. auris* cell wall...” and end with “The unique structure of chitin-bound β -1,3-glucan was consistently observed in both apo and caspofungin-treated *C. auris* cell walls. Therefore, the tight association between these two molecules in the rigid core represents a conserved structural feature that remains unaffected by echinocandin treatment, which is crucial for the mechanical strength of the cell wall and fungal survival.” We hope this can help to improve the significance of the analysis and results.

The Discussion section is excellent. It reviews the extensive number of important observations that lay the groundwork for eventually pinpointing new targets for antifungal drug discoveries. Subsequent experimental approaches based on these observations are explicitly proposed and are no doubt in progress.

We really appreciate the encouraging comments regarding the Discussion section.

The suggestions I have made may help make the paper more amenable to an audience versed in the biology of *Candida* species and drug development, but not well versed in the advanced technologies employed in the study and not well versed in the fungal wall literature. Most notably, I suggest considering diagrams of the cell wall that describe the inner and outer layers of the wall, that distinguish or define rigid scaffolds from mobile or flexible components, that distinguish hydrated from dehydrated components, that list the components and their characteristics in the inner and outer layers, and that describe the differences in the responses in the wall to caspofungin between *C. albicans*, *C. auris*, AR-386 and *C. auris*. I. 3, in the context of susceptibility, tolerance and resistance.

Yes, we greatly appreciate the helpful advice. In response, we have added a new Figure 8 to summarize the key concepts of *C. albicans* cell wall structure and integrate our new ssNMR findings into an updated model, informed by existing chemical analyses and imaging data from the literature. The figure includes four schematic representations for *C. albicans* and *C. auris* AR386 strains, both with and without caspofungin treatment, which are the primary focus of this study. For the I.3 strain, we indicate that no detectable changes in cell wall structure were observed regardless of caspofungin exposure. We hope this addition enhances the accessibility and clarity of our findings for a broad readership.

In summary, this is an excellent, in-depth study that will impact the field.

Thanks!

Reviewer #2

In this study, Widanage and colleagues investigated if the cell wall structures of *Candida auris* and *Candida albicans* were impacted by echinocandin treatment. The authors used complementary approaches, including solid-state NMR analyses. Overall, the authors found that the cell walls of both *Candida auris* and *Candida albicans* are similar. However, only *Candida albicans* responded by inducing cell wall thickening and changes in chitin and beta-1-3-glucan dynamics. This study provides a molecular-level understanding of the mechanisms that these pathogens use to overcome echinocandin treatment, which may be useful in future drug development studies.

We would like to thank the reviewer for highlighting the importance of this study.

My comments and suggestions are below:

SC5314 and AR386 have similar cell wall changes after caspofungin treatment, but I.3 is the most different. Can you expand on what contributes to the difference of I.3 and AR386 cell walls although they are both *C. auris*?

We have now added an explanation in the main text when discussing the results of Fig. 3: “None of these changes were observed in the *C. auris* I.3 strain, which exhibits echinocandin resistance due to the S639F mutation in the β -1,3-glucan synthase FKS1. The varying levels of echinocandin susceptibility among different strains may influence their responses to caspofungin challenge.”

Consider providing evidence or past studies that support that the biradical used for DNP does not change the cell wall architecture.

We have now added an explanation stating that “AsymPolPOK and other biradicals are small (2-3 nm), water-soluble molecules that diffuse through porous cell walls without disrupting their structure, enabling widespread use in the structural analysis of plant, fungal, algal, and bacterial cell wall materials as well as intact mammalian cells. Sensitivity enhancement is achieved by transferring electron spin polarization to ^1H nuclei in the biradical, followed by ^1H spin diffusion across the solvent and finally to biomolecules, resulting in homogeneous hyperpolarization over distances of at least tens of nanometers.” Additionally, seven new references have been included to support DNP studies on cell walls and intact cells across diverse organisms.

How long are the NMR experiments (this information was only provided for some experiments)? This is an important consideration in case the cells are changing over the course of very long experiments. Consider providing evidence that the cell wall compositions and architectures are the same at the beginning and end of the NMR experiments.

We have now included detailed information for each experiment in **Supplementary Tables 9 and 10**. In addition, a new **Supplementary Figure 13** has been provided to demonstrate the reproducibility of the carbohydrate regions in *Candida* samples over time. Samples are stored in a $-20\text{ }^\circ\text{C}$ freezer when not being measured. All 1D experiments are closely monitored before and after 2D/3D acquisitions to ensure sample integrity. Their spectra are highly reproducible. We have also added clarification to the **Methods** section.

Are the cells viable and growing normally after the solid-state NMR experiments? This would lend support to that the cell wall architecture, including levels of hydration, are not impacted during the solid-state NMR experiments and that the spectra that you report reflect native conditions.

Yes, the cells remain largely viable and can be used to initiate new cultures. We have now included this information in the **Methods** section and provided a new **Supplementary Figure 14** to support this observation.

beta-1,6-glucans and beta-1,3-glucans can compensate for one another. Which Kre6 proteins compensate for the similar or higher levels of beta-1,6-glucans? Can you provide more clarifications with citations?

KRE6A compensate for a similar level of β -1,6-glucan and KRE6b compensate for a higher level of β -1,6-glucan. We have expanded these two sentences by incorporating specific numbers and mentioning specific strains: “Exposure to caspofungin further altered the composition of the mobile matrix by increasing β -1,6-glucans, from 19% to 34% for *kre6a* Δ and from 23% to 53% for *kre6b* Δ , while reducing the content of β -1,3-glucan within each deletion mutant (Fig. 6g, right). These findings suggested that β -1,6-glucan and β -1,3-glucan have partially interchangeable roles in forming the mobile matrix, likely compensating for the loss of each other. Interestingly, the addition of caspofungin led to similar or even higher β -1,6-glucan levels in the single deletion mutants compared to AR386: the β -1,6-glucan level in the mobile fraction of caspofungin-treated cell walls was 35%, 34%, and 53% for wild-type, *kre6a* Δ , and *kre6b* Δ samples, respectively (Fig. 6g, right). Therefore, under caspofungin stress, a single KRE6 protein, either KRE6a or KRE6b (with KRE6b playing a more prominent role), is sufficient to support β -1,6-glucan biosynthesis.”

Consider including a brief statement describing your choice of CORD (example: Figure 1) instead of another similar pulse sequence such as PDS or DARR.

We have now clarified this point in the Methods section: “For *Candida* cells, the CORD experiment was performed at ambient temperature for resonance assignment. CORD efficiently drives homonuclear ^{13}C - ^{13}C polarization transfer under fast MAS conditions and high magnetic fields. At the slower MAS rate of 13.5 kHz used in this study, its performance is comparable to that of Dipolar-Assisted Rotational Resonance (DARR), while offering the advantage of lower radiofrequency power requirements during the recoupling period. Compared to proton-driven spin diffusion (PDS), CORD provides higher efficiency due to the use of active recoupling pulses.”

There seems to be a typo in the last sentence of the Figure 2 legend about the experimental parameters for the hcCH and hCH spectra.

We have corrected it to “The hcCH spectra were acquired on 800 MHz spectrometer under 13.5 kHz MAS, while the hCH spectra were measured on a 600 MHz spectrometer at 60 kHz MAS.”

There seems to be a typo in line 315, since I believe that you are referencing Figure 4A (not Fig 3A).

Yes, we have now corrected it to “Fig. 4a.” Thanks for catching the typo.

Reviewer #3:

This manuscript by Widanage, Singh, and Li et al. focuses on a detailed analysis of the cell wall structure of two fungal pathogens, *Candida albicans* and *Candida auris*. The analysis is similar to that performed for other fungal pathogens in response to stress, including *Aspergillus*, *Cryptococcus*, *Schizophyllum*, *Neurospora*, and *Mucor*. The major finding is that the *Candida* species characterized share a similar cell wall composition that is distinct from other fungi and show changes in response to echinocandin exposure. Although the cell wall analysis may present an incremental advance, there

are a number of issues with the manuscript. First, the stated goal of the manuscript of asking how echinocandins remodel cell wall structures, thereby reducing drug effectiveness is very strange. The echinocandins inhibit a key cell wall linker molecule as part of their mode of action and the impact on the cell wall is the basis of their effectiveness. The premise upon which it is inferred that cell wall changes induced by the echinocandins are the basis of reduced echinocandin efficacy is entirely unclear. Second, the conclusion that profound alterations induced by caspofungin in *Candida* cell wall architecture suggest that cell wall structural contribute substantially to drug resistance and tolerance is not supported by the data presented. A observation of cell wall changes accompanying drug treatment does not implicate cell wall changes in resistance. Additional issues are addressed in the specific points below.

We appreciate the critical comments of our reviewer. Below are the detailed responses regarding the specific comments, where we have generated new complement strains of the *kre6* mutants, compared different *C. albicans* strains, and modified the writing following the reviewer's advice. We have also removed the word "tolerance" following the advice.

Regarding the cell wall analysis, we would like to clarify that the large set of various methods are not uniformly applied across all fungal species. Rather, we are continuously developing and refining a methodological toolbox, with capabilities that evolve year by year. For instance, the ¹H data in the present work is of particularly high quality, and the DNP results for distinguishing the different functional structures of the same polysaccharide are notably unique; however, we would like to avoid emphasizing these technical aspects in order to maintain the focus of the study for the broader audience. Accordingly, we have rephrased the sentence in the Introduction to read: ". Some of these solid-state NMR techniques have been selectively applied and tailored—depending on the purpose of each study—to investigations of....."

Regarding the two comments about the abstract, we have now tuned down the statements. For the first point, it has been modified as "To understand how echinocandins remodel *Candida* cell wall structures, this study compares the effects of echinocandin exposure on the cell walls of..." For the second point, it has been modified as "These findings have resolved the distinct roles of various polysaccharides in constructing *Candida* cell walls and remodeling their architecture to support adaptative survival during antifungal exposure."

In addition, we have included a new Figure 8 to summarize the novel structural insights into *Candida* cell walls. We hope these changes can make this study more accessible to our readers.

Specific Points:

1. The background that echinocandin tolerance arises through the growth of subpopulations of cells that survive despite drug-induced stress cites two references, one of which focuses on azoles not echinocandins and the other is a broad review article. This point needs to be appropriately supported by the literature.

We agree with the reviewer that the term “tolerance” might be not appropriate regarding the effect of echinocandins against *Candida*. The goal of the study is not to investigate echinocandin tolerance, but rather the *Candida* response to echinocandin treatment. Therefore, we have removed the term “tolerance” throughout the manuscript.

2. If the goal of the study is to understand tolerance, then there should be analysis of isogenic cells that acquired tolerance relative to their sensitive parental strains.

We have removed the term “tolerance” throughout the manuscript to make the study more focused.

3. For the comparative analysis shown in Figure 1, were the *Aspergillus* cells walls prepared in the same way as the *Candida* cells walls? If not, could the methodology affect the results? Were the *Aspergillus* results previously published (in which case this would need to be addressed).

We appreciate this critique, which aligns with the feedback from Reviewer 1. In response, we have removed the original Fig. 1a and 1c, as well as the *A. fumigatus* components from the original Fig. 1d and 1f. Additionally, we have fully rewritten the first subsection of the Results to focus specifically on the rigid and dynamic molecules in *C. albicans*, highlighting their structural roles within the cell wall.

4. Only one *Candida albicans* strain is used as representative of the entire species, despite large phenotypic variation well documented across isolates.

We have now added a new **Supplementary Figure 1** to demonstrate that the distribution of dynamically distinct molecules is consistently observed across multiple *C. albicans* strains.

5. There is no rationale provided for the specific concentration of echinocandin used for each species. Different effects could be attributable to different effective concentrations of echinocandins with no analysis at different doses. If the dose of caspofungin has minimal antifungal effect, as with *Candida auris*, it would be consistent that there is minimal effects on the cell wall.

We chose the concentration to be well below the MIC in order to collect enough material for NMR analysis, while simultaneously observing enough structural changes. The concentration (SC5314: 0.015 µg/mL; AR386: 0.05 µg/mL) used for *C. albicans* SC5314 and *C. auris* AR386 strain are facing a comparable level of challenge from caspofungin, allowing for maintaining a comparable level (90%) of growth for both strains, which will also be sufficient for NMR analysis as these two strains are the focus of this study. For the resistant I.3 strain of *C. auris*, although the drug concentration is five-fold higher than for AR386 and more than 30-fold higher than SC5314, it can maintain a high level of growth. We have now clarified this point in the maintext.

6. To interpret the phenotype of a gene deletion mutant and rule out the possibility of phenotypic effects being attributable to spurious mutations, it is critical to generate a complemented strain.

Thanks for the advice and we fully agree. We have generated the complemented strains of the two deletions strains (*kre6aΔ::KRE6a* and *kre6bΔ::KRE6b*). See the new **Figure 6a, 6c, 6d, and 6e**, as well as the **Supplementary Figures 10-12**. We also fully rewrite the subsection of “β-1,6-glucans modulate the susceptibility of *C. auris* to echinocandins” to include these new results.

7. The authors conclude FKS1 is overexpressed in the *kre6a* mutant but the data shown is questionable – the magnitude of effect is very small and unlikely to contribute any increase in resistance.

The change in *FKSI* expression in *kre6aΔ* is modest (1.2-fold), but this upregulation was consistently observed and statistically significant across three independent experiments. Moreover, the complemented strain restored the reduced echinocandin susceptibility and reversed the *FKSI* expression change. We agree that it might be surprising that this modest upregulation of *FKSI* would be sufficient to alter echinocandin susceptibility. However, the decrease of echinocandin susceptibility was also very modest. We have added one sentence in the results section to comment on this result: “Although this increase of *FKSI* expression was modest (1.2-fold), it could be sufficient to induce the slight decrease of echinocandin susceptibility of this strain.”

Minor Points

Lines 604-606: This statement needs to be refined as not all *Candida* species transition from yeast to filamentous growth

Thanks. We have now modified this sentence as “Certain *Candida* species can transition from yeast forms to filamentous structures, producing true hyphae in species like *C. albicans*, *C. dubliniensis*, and *C. tropicalis*, or *pseudohyphae* in species such as *C. auris* and *C. parapsilosis*. The pathogenicity and immunogenicity of *Candida* species are closely linked to these morphological alterations”

Line 26: implicating not the correct word

Thanks. We have changed it to “resulting in”

Line 63: Echinocandin should not be capitalized mid sentence

Thanks for catching this typo. It should be a full stop before this word. We have now corrected it.

Line 65-66: “However, echinocandin treatment often results only in partial inhibition of β-1,3-glucan synthesis” Requires a reference.

Thanks. We have included key references.

Responses to Reviewers' Comments

Reviewer #1:

Review of revisions by Widanage et al.

The authors have revised the manuscript in response to all of the suggestions I articulated in the original review. These changes are outlined below.

1. As I suggested, the *A. fumigatus* results were interesting but served no purpose in comparative analysis of *Candida* species and the basis of the drug susceptibility differences between the latter. As recommended, the authors removed these results. This included removal of Figures 1a, c, d and f, Table 1, supplemental Figure 1, and the first section of Results.
2. I suggested clarifying the molecular differences between rigid and mobile layers of the cell wall. This has been dealt with adequately in the first section of Results.
3. The authors have added, as suggested, p-values for the differences between apo and drug tested drug-treated samples, demonstrating significance.
4. As suggested, the authors have adequately explained the relevance of their observation of the consistent structure of beta-1, 3-glucan bound by chitin in the apo and caspofungin-treated cell wall of *C. auris* in original lines 381-383.
5. Finally, the authors have added, as suggested, a new Figure 8 which diagrams for both *Candida* species the cell wall without and with caspofungin.

I might also point out that the authors analyzed complemented strains of the KRE mutants in response to the suggestion by reviewer 3, a point I had considered, but forgot to add to my original review.

I commend the authors for attending to the revisions and suggestions rather than rebutting the suggestions made by the three reviewers.

The editorial board may name me if the paper is published.

David R. Soll
Carver/Witschi Professor, Emeritus
University of Iowa

Response: We sincerely appreciate the reviewer's valuable guidance, which significantly contributed to the improvement of our manuscript during the previous revision. We are grateful for the reviewer's support for its publication.

Reviewer #2 (Remarks to the Author):

Thank you for addressing my prior comments. I have no additional comments, suggestions, or concerns.

Response: We deeply appreciate the support from the reviewer for our NMR study on *Candida* species.

Reviewer #3 (Remarks to the Author):

The authors have sufficiently addressed all my original critiques and I now recommend this manuscript be accepted to Nature Communications.

Response: Thanks! We hope that the current version will offer meaningful insights to the research community.